# Collaborative Score Distillation for Consistent Visual Editing

**Subin Kim**[*,1]     **Kyungmin Lee**[*,1]     **June Suk Choi**[1]     **Jongheon Jeong**[1]

**Kihyuk Sohn**[2]     **Jinwoo Shin**[1]

[1]KAIST     [2]Google Research

[*]{subin-kim, kyungmnlee}@kaist.ac.kr

## Abstract

Generative priors of large-scale text-to-image diffusion models enable a wide range of new generation and editing applications on diverse visual modalities. However, when adapting these priors to complex visual modalities, often represented as multiple images (e.g., video or 3D scene), achieving consistency across a set of images is challenging. In this paper, we address this challenge with a novel method, Collaborative Score Distillation (CSD). CSD is based on the Stein Variational Gradient Descent (SVGD). Specifically, we propose to consider multiple samples as "particles" in the SVGD update and combine their score functions to distill generative priors over a set of images synchronously. Thus, CSD facilitates the seamless integration of information across 2D images, leading to a consistent visual synthesis across multiple samples. We show the effectiveness of CSD in a variety of editing tasks, encompassing the visual editing of panorama images, videos, and 3D scenes. Our results underline the competency of CSD as a versatile method for enhancing inter-sample consistency, thereby broadening the applicability of text-to-image diffusion models.[1]

## 1 Introduction

Text-to-image diffusion models [1, 2, 3, 4] have been scaled up by using billions of image-text pairs [5, 6] and efficient architectures [7, 8, 9, 4], showing impressive capability in synthesizing high-quality, realistic, and diverse images with the text given as an input. Furthermore, they have branched into various applications, such as image-to-image translation [10, 11, 12, 13, 14, 15, 16], controllable generation [17], or personalization [18, 19]. One of the latest applications in this regard is to translate the capability into other complex modalities, viz., beyond 2D images [20, 21] without modifying diffusion models using modality-specific training data. This paper focuses on the problem of adapting the knowledge of pre-trained text-to-image diffusion models to more complex high-dimensional visual manipulation tasks beyond 2D images without modifying diffusion models using modality-specific training data.

We start from an intuition that many complex visual data, e.g., videos and 3D scenes, are represented as a *set of images* constrained by modality-specific consistency. For example, a video is a set of frames requiring temporal consistency, and a 3D scene is a set of multi-view frames with view consistency. Unfortunately, image diffusion models do not have a built-in capability to ensure consistency between a set of images for synthesis or editing because their generative sampling process does not take into account the consistency when using the image diffusion model as is. As such, when applying image diffusion models to manipulate these complex data without consistency in consideration, it

---

[*]Equal contribution.

[1]Visualizations are available at the website https://subin-kim-cv.github.io/CSD.

37th Conference on Neural Information Processing Systems (NeurIPS 2023).

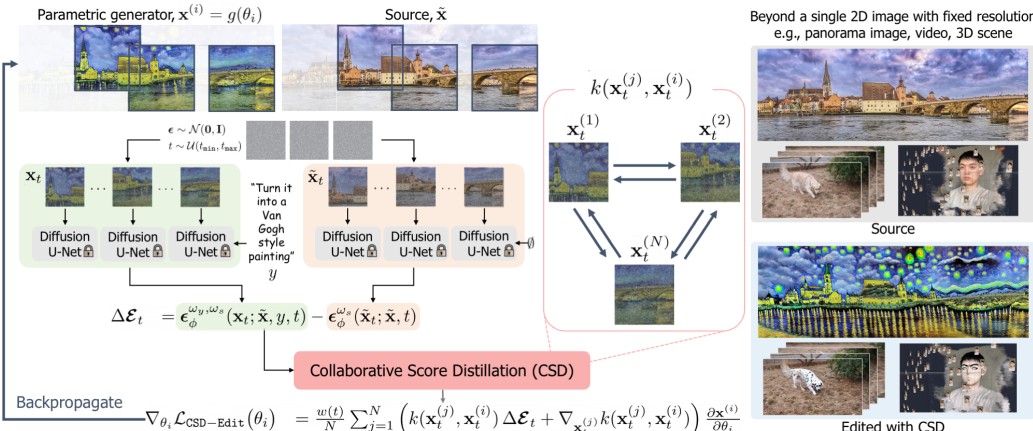

Figure 1: **Method overview**. CSD-Edit enables various visual-to-visual translations with two novel components. First, a new score distillation scheme using Stein variational gradient descent, which considers inter-sample relationships (Section 3.1) to synthesize a set of images while preserving modality-specific consistency constraints. Second, our method edits images with minimal information given from text instruction by subtracting image-conditional noise estimate instead of random noise during score distillation (Section 3.2). By doing so, CSD-Edit is used for text-guided manipulation of various visual domains, e.g., panorama images, videos, and 3D scenes (Section 3.3).

results in a highly incoherent output, as in Figure 2 (Patch-wise Crop), where one can easily identify where images are being stitched. Such behaviors are also reported in video editing, thus, recent works [22, 23, 24, 25] propose to handle video-specific temporal consistency when using the image diffusion model.

Here, we take attention to an alternative approach, Score Distillation Sampling (SDS) [26], which enables the optimization of arbitrary differentiable operators by leveraging the rich generative prior of text-to-image diffusion models. SDS poses generative sampling as an optimization problem by distilling the learned diffusion density scores. While Poole et al. [26] has shown the effectiveness of SDS in generating 3D objects from the text by resorting on Neural Radience Fields [27] priors which inherently suppose coherent geometry in 3D space through density modeling, it has not been studied for consistent visual manipulation of other modalities, where modality-specific consistency constraints should be considered when manipulating.

In this paper, we propose *Collaborative Score Distillation* (CSD), a simple yet effective method that extends the singular of the text-to-image diffusion model for consistent visual manipulation. The crux of our method is two-fold: first, we establish a generalization of SDS by using Stein variational gradient descent (SVGD), where multiple samples share their knowledge distilled from diffusion models to accomplish inter-sample consistency. Second, we present CSD-Edit, an effective method for consistent visual editing by leveraging CSD with Instruct-Pix2Pix [14], a recently proposed instruction-guided image diffusion model (See Figure 1).

We demonstrate the versatility of our method in various editing applications such as panorama image editing, video editing, and reconstructed 3D scene editing. In editing a panorama image, we show that CSD-Edit obtains spatially consistent image editing by optimizing multiple patches of an image. Also, compared to other methods, our approach achieves a better trade-off between source-target image consistency and instruction fidelity. In video editing experiments, CSD-Edit obtains temporal consistency by taking multiple frames into optimization, resulting in temporal frame-consistent video editing. Furthermore, we apply CSD-Edit to 3D scene editing and generation, by encouraging consistent manipulation and synthesis among multiple views.

## 2 Preliminaries

### 2.1 Diffusion models

Generative modeling with diffusion models consists of a forward process $q$ that gradually adds Gaussian noise to the input $\mathbf{x}_0 \sim p_{\texttt{data}}(\mathbf{x})$, and a reverse process $p$ which gradually denoises from

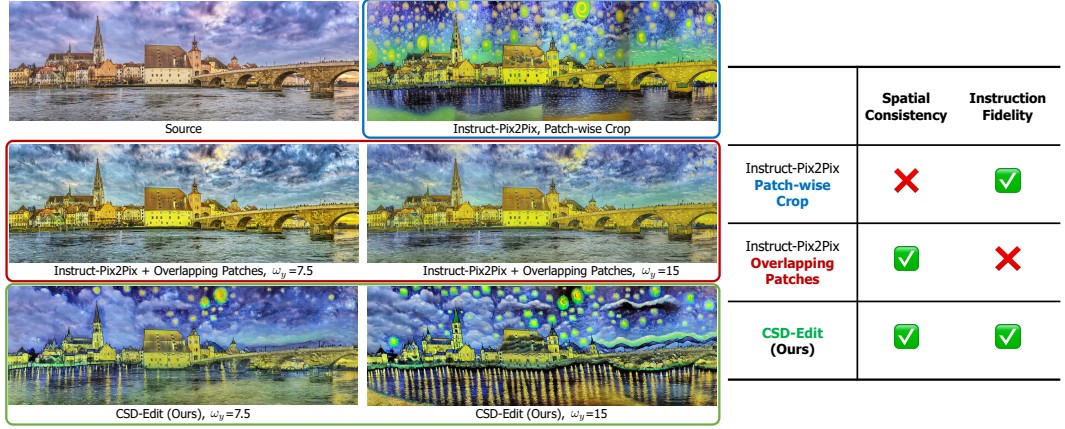

Figure 2: **Panorama image editing**. (Top right) Instruct-Pix2Pix [14] on cropped patches results in inconsistent image edits. (Second row) Instruct-Pix2Pix on overlapping patches edits to a consistent image, but less fidelity to the instruction, even with high guidance scale $\omega_y$. (Third row) CSD-Edit provides consistent image editing with better instruction-fidelity by setting a proper guidance scale.

the Gaussian noise $\mathbf{x}_T \sim \mathcal{N}(\mathbf{0}, \mathbf{I})$. Formally, the forward process $q(\mathbf{x}_t|\mathbf{x}_0)$ at timestep $t$ is given by $q(\mathbf{x}_t|\mathbf{x}_0) = \mathcal{N}(\mathbf{x}_t; \alpha_t\mathbf{x}_0, \sigma_t^2\mathbf{I})$, where $\sigma_t$ and $\alpha_t^2 = 1 - \sigma_t^2$ are pre-defined constants designed for effective modeling [8, 28, 29]. Given enough timesteps, reverse process $p$ also becomes a Gaussian and the transitions are given by posterior $q$ with optimal MSE denoiser [30], i.e., $p_\phi(\mathbf{x}_{t-1}|\mathbf{x}_t) = \mathcal{N}(\mathbf{x}_{t-1}; \mathbf{x}_t - \hat{\mathbf{x}}_\phi(\mathbf{x}_t; t), \sigma_t^2\mathbf{I})$, where $\hat{\mathbf{x}}_\phi(\mathbf{x}_t; t)$ is a learned optimal MSE denoiser. Ho et al. [7] proposed to train an U-Net [31] autoencoder $\boldsymbol{\epsilon}_\phi(\mathbf{x}_t; t)$ by minimizing following objective:

$$\mathcal{L}_{\text{Diff}}(\phi; \mathbf{x}) = \mathbb{E}_{t\sim\mathcal{U}(0,1), \boldsymbol{\epsilon}\sim\mathcal{N}(\mathbf{0},\mathbf{I})}\big[w(t)\|\boldsymbol{\epsilon}_\phi(\mathbf{x}_t; t) - \boldsymbol{\epsilon}\|_2^2\big], \quad \mathbf{x}_t = \alpha_t\mathbf{x}_0 + \sigma_t\boldsymbol{\epsilon} \tag{1}$$

where $w(t)$ is a weighting function for each timestep $t$. Text-to-image diffusion models [1, 2, 4, 3] are trained by Eq. (1) with $\boldsymbol{\epsilon}_\phi(\mathbf{x}_t; y, t)$ that estimates the noise conditioned on the text prompt $y$. To effectively guide the text-conditional generation, Ho et al. [32] proposed classifier-free guidance (CFG), where they jointly train the unconditional and conditional model and interpolate the unconditional and conditional model during the inference, *i.e.*, the noise estimate is given by

$$\boldsymbol{\epsilon}_\phi^\omega(\mathbf{x}_t; y, t) = \boldsymbol{\epsilon}_\phi(\mathbf{x}_t; t) + \omega_y\big(\boldsymbol{\epsilon}_\phi(\mathbf{x}_t; y, t) - \boldsymbol{\epsilon}_\phi(\mathbf{x}_t; t)\big), \tag{2}$$

where $\omega_y \geq 0$ is a guidance scale that controls the sample fidelity. Specifically, increasing $\omega_y$ enhances sample fidelity at the expense of sample diversity. Throughout the paper, we refer $p_\phi^{\omega_y}(\mathbf{x}_t; y, t)$ a conditional distribution of a text $y$.

**Instruction-based image editing by Instruct-Pix2Pix.** Recently, many works have demonstrated the capability of diffusion models in editing or stylizing images [10, 13, 11, 12, 14]. Among them, Brooks et al. [14] proposed Instruct-Pix2Pix, where they finetuned Stable Diffusion [4] model with the source image, text instruction, (edited) target image (edited by Prompt-to-Prompt [12]) triplets to enable instruction-based editing of an image. Then during the inference, Instruct-Pix2Pix starts from a source image and conducts diffusion sampling by the diffusion model that takes instruction $y$. In specific, given the source image $\tilde{\mathbf{x}}$ and instruction $y$, the noise estimate at time $t$ is given by

$$\begin{aligned}\boldsymbol{\epsilon}_\phi^{\omega_s, \omega_y}(\mathbf{x}_t; \tilde{\mathbf{x}}, y, t) = \boldsymbol{\epsilon}_\phi(\mathbf{x}_t; t) &+ \omega_s\big(\boldsymbol{\epsilon}_\phi(\mathbf{x}_t; \tilde{\mathbf{x}}, t) - \boldsymbol{\epsilon}_\phi(\mathbf{x}_t; t)\big) \\ &+ \omega_y\big(\boldsymbol{\epsilon}_\phi(\mathbf{x}_t; \tilde{\mathbf{x}}, y, t) - \boldsymbol{\epsilon}_\phi(\mathbf{x}_t; \tilde{\mathbf{x}}, t)\big),\end{aligned} \tag{3}$$

where $\omega_y \geq 0$ is the CFG parameter for text instruction as in Eq. (2) and $\omega_s \geq 0$ is an additional CFG parameter that controls the fidelity to the source image $\tilde{\mathbf{x}}$.

## 2.2 Score distillation sampling

Poole et al. [26] proposed Score Distillation Sampling (SDS), an alternative sample generation method by distilling the rich knowledge of text-to-image diffusion models. SDS allows optimization of any differentiable image generator, e.g., Neural Radiance Fields [27] or the image space itself.

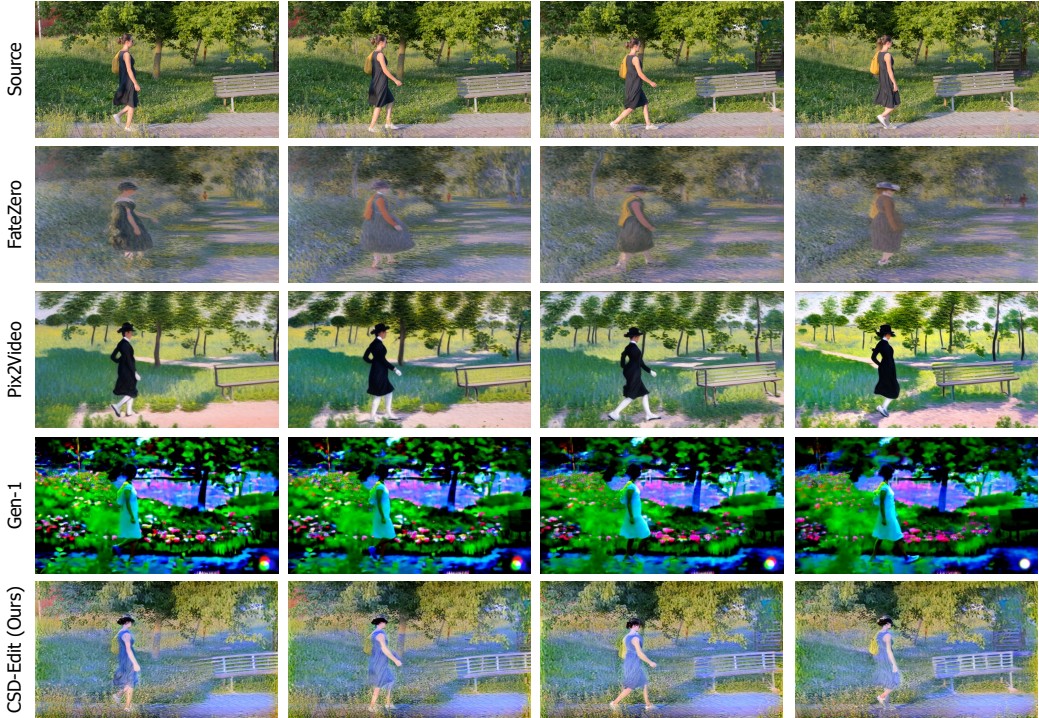

*"Make it as a painting of Claude Monet"*

Figure 3: **Video editing**. Qualitative results on the lucia video in DAVIS 2017 [33]. CSD shows frame-wise consistent editing providing coherent content across video frames e.g., consistent color and background without changes in person. Compared to Gen-1 [21], a video editing method trained on a large video dataset, CSD-Edit shows high-quality video editing results reflecting given prompts.

Formally, let $\mathbf{x} = g(\theta)$ be an image rendered by a differentiable generator $g$ with parameter $\theta$, then SDS minimizes density distillation loss [34] which is a variational inference via minimizing KL divergence between the posterior of $\mathbf{x} = g(\theta)$ and the text-conditional density $p_\phi^\omega$:

$$\min_\theta \mathcal{L}_{\texttt{Distill}}\big(\theta; \mathbf{x} = g(\theta)\big) = \mathbb{E}_{t,\boldsymbol{\epsilon}}\big[\alpha_t/\sigma_t\, D_{\text{KL}}\big(q\big(\mathbf{x}_t|\mathbf{x} = g(\theta)\big) \,\|\, p_\phi^\omega(\mathbf{x}_t; y, t)\big)\big], \qquad (4)$$

where $\mathbf{x}_t = \alpha_t \mathbf{x} + \sigma_t \boldsymbol{\epsilon}$ with $\mathbf{x} = g(\theta)$ and $\boldsymbol{\epsilon} \sim \mathcal{N}(\mathbf{0}, \mathbf{I})$. They derive SDS by differentiating Eq. 4 with respect to generator parameter $\theta$, but omitting the U-Net Jacboian term due to its poor performance and computationally inefficient. The SDS gradient update is given as follows:

$$\nabla_\theta \mathcal{L}_{\text{SDS}}\big(\theta; \mathbf{x} = g(\theta)\big) = \mathbb{E}_{t,\boldsymbol{\epsilon}}\left[w(t)\big(\boldsymbol{\epsilon}_\phi^\omega(\mathbf{x}_t; y, t) - \boldsymbol{\epsilon}\big)\frac{\partial \mathbf{x}}{\partial \theta}\right]. \qquad (5)$$

In its implementation, we randomly sample timestep from uniformly distributed interval $\mathcal{U}[t_{\texttt{min}}, t_{\texttt{max}}]$. The range of timesteps $t_{\texttt{min}}$ and $t_{\texttt{max}}$ are chosen to sample from not too small or large noise levels, and the guidance scales are chosen to be larger than those used for image generation.

## 2.3 Stein variational gradient descent

The original motivation of Stein variational gradient descent (SVGD) [35] is to solve a variational inference problem, where the goal is to approximate a target distribution from a simpler distribution by minimizing KL divergence. Formally, suppose $p$ is a target distribution with a known score function $\nabla_{\mathbf{x}} \log p(\mathbf{x})$ that we aim to approximate, and $q(\mathbf{x})$ is a known source distribution. Liu and Wang [35] showed that the steepest descent of KL divergence between $q$ and $p$ is given as follows:

$$\mathbb{E}_{q(\mathbf{x})}\big[\mathbf{f}(\mathbf{x})^\top \nabla_{\mathbf{x}} \log p(\mathbf{x}) + \text{Tr}(\nabla_{\mathbf{x}}\mathbf{f}(\mathbf{x}))\big], \qquad (6)$$

where $\mathbf{f} : \mathbb{R}^D \to \mathbb{R}^D$ is any smooth vector function that satisfies $\lim_{\|\mathbf{x}\|\to\infty} p(\mathbf{x})\mathbf{f}(\mathbf{x}) = 0$. Remark that Eq. (6) becomes zero if we replace $q(\mathbf{x})$ with $p(\mathbf{x})$ in the expectation term, which is known as

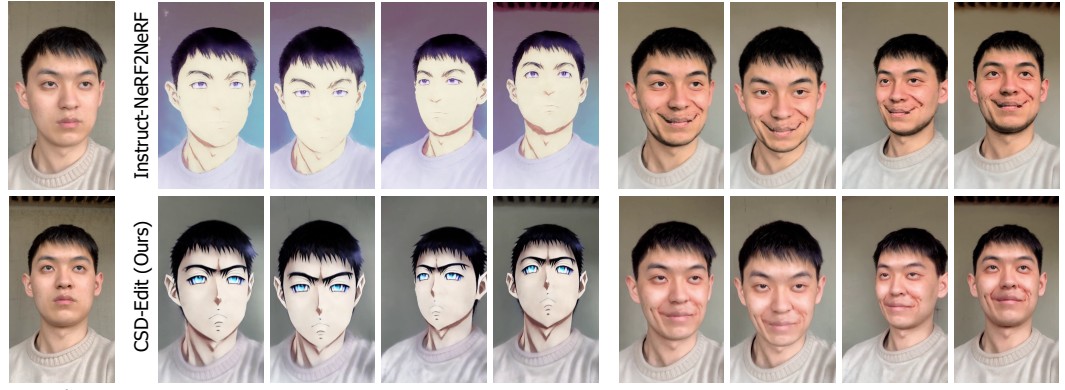

Figure 4: **3D NeRF scene editing**. Visualizing novel-views of edited Fangzhou NeRF scene [37]. CSD-Edit leads to high-quality editing of 3D scenes and better preserves semantics of source scenes, e.g., obtains sharp facial details (left) and makes him smile without giving beard (right).

Stein's identity [36]. Here, the choice of the critic **f** is crucial in its convergence and computational tractability. To that end, Liu and Wang [35] proposed to constrain **f** in the Reproducing Kernel Hilbert Space (RKHS) which yields a closed-form solution. Specifically, given a positive definite kernel $k : \mathbb{R}^D \times \mathbb{R}^D \to \mathbb{R}^+$, Stein variational gradient descent provides the greedy directions as follows:

$$\mathbf{x} \leftarrow \mathbf{x} - \eta \Delta\mathbf{x}, \quad \Delta\mathbf{x} = \mathbb{E}_{q(\mathbf{x}')}\big[k(\mathbf{x}, \mathbf{x}')\nabla_{\mathbf{x}'} \log p(\mathbf{x}') + \nabla_{\mathbf{x}'} k(\mathbf{x}, \mathbf{x}')\big], \quad (7)$$

with small step size $\eta > 0$. The SVGD update in Eq. (7) consists of two terms that play different roles: the first term moves the particles towards the high-density region of target density $p(\mathbf{x})$, where the direction is smoothed by kernels of other particles. The second term acts as a repulsive force that prevents the mode collapse of particles. One can choose different kernel functions, while we resort to standard Radial Basis Function (RBF) kernel $k(\mathbf{x}, \mathbf{x}') = \exp(-\frac{1}{h}\|\mathbf{x} - \mathbf{x}'\|_2^2)$ with bandwidth $h > 0$.

## 3 Method

In this section, we introduce *Collaborative Score Distillation* (CSD) for consistent synthesis and editing of multiple samples. We first derive a collaborative score distillation method using Stein variational gradient descent (Section 3.1) and propose an effective image editing method using CSD, i.e., CSD-Edit, that leads to coherent editing of multiple images with instruction (Section 3.2). Lastly, we present various applications of CSD-Edit in editing panorama images, videos, and 3D scenes (Section 3.3).

### 3.1 Collaborative score distillation

Suppose a set of parameters $\{\theta_i\}_{i=1}^N$ that generates images $\mathbf{x}^{(i)} = g(\theta_i)$. Similar to SDS, our goal is to update each $\theta_i$ by distilling the smoothed densities from the diffusion model by minimizing KL divergence in Eq. (4). On the contrary, CSD solves Eq. (4) using SVGD demonstrated in Section 2.3 so that each $\theta_i$ can be updated in sync with updates of other parameters in the set $\{\theta_i\}_{i=1}^N$. At each update, CSD samples $t \sim \mathcal{U}(t_{\texttt{min}}, t_{\texttt{max}})$ and $\boldsymbol{\epsilon} \sim \mathcal{N}(\mathbf{0}, \mathbf{I})$, and update each $\theta_i$ as follows:

$$\nabla_{\theta_i} \mathcal{L}_{\text{CSD}}(\theta_i) = \frac{w(t)}{N} \sum_{j=1}^N \Big( k(\mathbf{x}_t^{(j)}, \mathbf{x}_t^{(i)})(\boldsymbol{\epsilon}_\phi^\omega(\mathbf{x}_t^{(j)}; y, t) - \boldsymbol{\epsilon}) + \nabla_{\mathbf{x}_t^{(j)}} k(\mathbf{x}_t^{(j)}, \mathbf{x}_t^{(i)}) \Big) \frac{\partial \mathbf{x}^{(i)}}{\partial \theta_i}, \quad (8)$$

for each $i = 1, 2, \dots, N$. We refer to Appendix A for full derivation. Note CSD is equivalent to SDS in Eq. (5) when $N = 1$, showing that CSD is a generalization of SDS to multiple samples. As the pairwise kernel values are multiplied by the noise prediction term, each parameter update on $\theta_i$ is affected by other parameters, i.e., the scores are mixed with importance weights according to the affinity among samples. The more similar samples tend to exchange more score updates, while different samples tend to interchange the score information less. The gradient of the kernels acts as a repulsive force that prevents the mode collapse of samples. Moreover, we note that Eq. (8) does not

make any assumption on the relation between $\theta_i$'s or their order besides them being a set of images to be synthesized coherently with each other. As such, CSD is also applicable to arbitrary image generators, as well as text-to-3D synthesis in DreamFusion [26], which we compare in Section 4.4.

## 3.2 Instruction-guided editing by collaborative score distillation

In this section, we introduce an instruction-guided visual editing method using Collaborative Score Distillation (CSD-Edit). Given source images $\tilde{\mathbf{x}}^{(i)} = g(\tilde{\theta}_i)$ with parameters $\tilde{\theta}_i$, we optimize new target parameters $\{\theta_i\}_{i=1}^N$ with $\mathbf{x}^{(i)} = g(\theta_i)$ such that 1) each $\mathbf{x}^{(i)}$ follows the instruction prompt, 2) preserves the semantics of source images as much as possible, and 3) the obtained images are consistent with each other. To this end, we update each parameter $\theta_i$, initialized with $\tilde{\theta}_i$, using CSD with noise estimate $\boldsymbol{\epsilon}_\phi^{\omega_y,\omega_s}$ of Instruct-Pix2Pix [14]. However, this approach often results in blurred outputs, leading to the loss of details of the source image (see Figure 6). This is because the score distillation term subtracts random noise $\epsilon$, which perturbs the undesirable details of source images.

We handle this issue by adjusting the noise prediction term that enhances the consistency between source and target images. Subtracting a random noise $\epsilon$ in Eq. (5) when computing the gradient is a crucial factor, which helps optimization by reducing the variance of a gradient. Therefore, we amend the optimization by changing the random noise into a better baseline function. Since our goal is to edit an image with only minimal information given text instructions, we set the baseline by the image-conditional noise estimate $\boldsymbol{\epsilon}_\phi^{\omega_s}$ of the Instruct-Pix2Pix model without giving text instructions on the source image. To be specific, our CSD-Edit is given as follows:

$$\nabla_{\theta_i} \mathcal{L}_{\texttt{CSD-Edit}}(\theta_i) = \frac{w(t)}{N} \sum_{j=1}^N \left( k(\mathbf{x}_t^{(j)}, \mathbf{x}_t^{(i)}) \Delta \boldsymbol{\mathcal{E}}_t^{(i)} + \nabla_{\mathbf{x}_t^{(j)}} k(\mathbf{x}_t^{(j)}, \mathbf{x}_t^{(i)}) \right) \frac{\partial \mathbf{x}^{(i)}}{\partial \theta_i}, \tag{9}$$

$$\Delta \boldsymbol{\mathcal{E}}_t^{(i)} = \boldsymbol{\epsilon}_\phi^{\omega_y,\omega_s}(\mathbf{x}_t^{(i)}; \tilde{\mathbf{x}}, y, t) - \boldsymbol{\epsilon}_\phi^{\omega_s}(\tilde{\mathbf{x}}_t^{(i)}; \tilde{\mathbf{x}}, t).$$

In Section 4.4, we validate our findings on the effect of baseline noise on image editing performance. We notice that CSD-Edit presents an alternative way to utilize Instruct-Pix2Pix in image-editing without any finetuning of diffusion models, by posing an optimization problem.

## 3.3 CSD-Edit for various complex visual domains

**Panorama image editing.** Diffusion models are usually trained on a fixed resolution (e.g., $512 \times 512$ for Stable Diffusion [4]), thus when editing a panorama image (i.e., an image with a large aspect ratio), the editing quality significantly degrades. Otherwise, one can crop an image into smaller patches and apply image editing on each patch. However this results in spatially inconsistent images (see Figure 2, Patch-wise Crop, Appendix D). To that end, we propose to apply CSD-Edit on patches to obtain spatially consistent editing of an image, while preserving the semantics of the source image. Following [38], we sample patches of size $512 \times 512$ that overlap using small stride and apply CSD-Edit on the latent space of Stable Diffusion [4]. Since we allow overlapping, some pixels might be updated more frequently. Thus, we normalize the gradient of each pixel by counting the appearance. Remark that one can give different instructions on the different regions of an image while maintaining consistency (See Appendix C for details).

**Video editing.** Editing a video with an instruction should satisfy the following: 1) temporal consistency between frames such that the degree of changes compared to the source video should be consistent across frames, 2) ensuring that desired edits in each edited frame are in line with the given prompts while preserving the original structure of source video, and 3) maintaining the sample quality in each frame after editing. To meet these requirements, we randomly sample a batch of frames and update them with CSD-Edit to achieve temporal consistency between frames.

**3D scene editing.** We consider editing a 3D scene reconstructed by a Neural Radiance Field (NeRF) [27], which represents volumetric 3D scenes using 2D images. To edit reconstructed 3D NeRF scenes, it is straightforward to update the training views with edited views and finetune the NeRF with edited views. Here, the multi-view consistency between edited views should be considered since inconsistencies between edits across multiple viewpoints lead to blurry and undesirable artifacts, hindering the optimization of NeRF. To mitigate this, Haque et al. [39] proposed

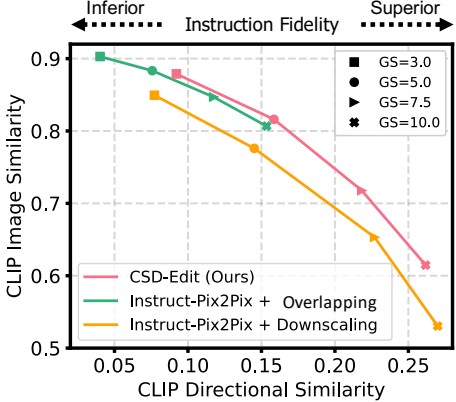

Figure 5: **Panorama image editing**. Comparison of CSD-Edit with baselines at different guidance scales $\omega_y \in \{3.0, 5.0, 7.5, 10.0\}$.

Table 1: **Video editing**. Quantitative comparison of CSD-Edit with baselines on video editing. Bold indicates the best results.

|  | CLIP Directional Similarity ↑ | CLIP Image Consistency ↑ | LPIPS ↓ |
|---|---|---|---|
| FateZero [22] | $0.312_{\pm 0.003}$ | $0.948_{\pm 0.001}$ | $0.264_{\pm 0.002}$ |
| Pix2Video [25] | $0.229_{\pm 0.001}$ | $0.948_{\pm 0.001}$ | $0.282_{\pm 0.001}$ |
| **CSD-Edit (Ours)** | $\mathbf{0.319_{\pm 0.002}}$ | $\mathbf{0.957_{\pm 0.001}}$ | $\mathbf{0.235_{\pm 0.001}}$ |

Table 2: **3D scene editing**. Quantitative comparison of CSD-Edit with baselines on 3D scene editing. Bold indicates the best results.

|  | CLIP Directional Similarity ↑ | CLIP Image Consistency ↑ | LPIPS ↓ |
|---|---|---|---|
| IN2N [14] | $0.177_{\pm 0.062}$ | $0.993_{\pm 0.002}$ | $0.053_{\pm 0.028}$ |
| **CSD-Edit (Ours)** | $\mathbf{0.215_{\pm 0.052}}$ | $\mathbf{0.994_{\pm 0.001}}$ | $\mathbf{0.045_{\pm 0.012}}$ |

Instruct-NeRF2NeRF, which performs editing on a subset of training views and updates them sequentially at training iteration with intervals. However, image-wise editing results in inconsistencies between views, thus they rely on the ability of NeRF in achieving multi-view consistency. Contrary to Instruct-NeRF2NeRF, we update the dataset with multiple consistent views through CSD-Edit, which serves as better training resources for NeRF, leading to less artifacts and better preservation of source 3D scene.

# 4 Experiments

## 4.1 Text-guided panorama image editing

For the panorama image-to-image translation task, we compare CSD-Edit with different versions of Instruct-Pix2Pix: one is which using naive downsizing to $512 \times 512$ and performing Instruct-Pix2Pix, and another is updating Instruct-Pix2Pix on the patches cropped with overlapping as in MultiDiffusion [38] (Instruct-Pix2Pix + Overlapping). For comparison, we collect a set of panorama images (i.e., which aspect ratio is higher than 3), and edit each image to various artistic styles and different guidance scales $\omega_y$. For evaluation, we use pre-trained CLIP [40] to measure two different metrics: 1) consistency between source and target images by computing similarity between two image embeddings, and 2) CLIP directional similarity [41] which measures how the change in text agrees with the change in the images. The experimental details are in Appendix B.1.

In Figure 5, we plot the CLIP scores of different image editing methods with different guidance scales. We notice that CSD-Edit provides the best trade-off between the consistency between source and target images and fidelity to the instructions. Figure 2 provides a qualitative comparison between panorama image editing methods. Note that applying Instruct-Pix2Pix to patches cropped with overlapping (Instruct-Pix2Pix + Overlapping) is able to edit images while preserving spatial consistency, as evidenced by high CLIP image similarity, however, the edited images show inferior fidelity to the text instruction even when using a large guidance scale, resulting in much lower CLIP directional similarity. We conjecture that this happens because the scores are diluted by other scores, i.e., one patch may respond much more or much less to the instruction compared to others. Additional qualitative results are in Appendix D.

## 4.2 Text-guided video editing

For the video editing experiments, we primarily compare CSD-Edit with existing zero-shot video editing schemes that employ text-to-image diffusion models such as FateZero [22] and Pix2Video [25]. To emphasize the effectiveness of CSD-Edit over learning-based schemes, we also compare it with Gen-1 [21], a state-of-the-art video editing method trained on a large video dataset. For quantitative evaluation, we report CLIP image-text directional similarity as in Section 4.1 to measure the alignment between changes in text and images. We also measure CLIP image consistency and LPIPS [42] between consecutive frames to evaluate temporal consistency. In addition to the objective metrics

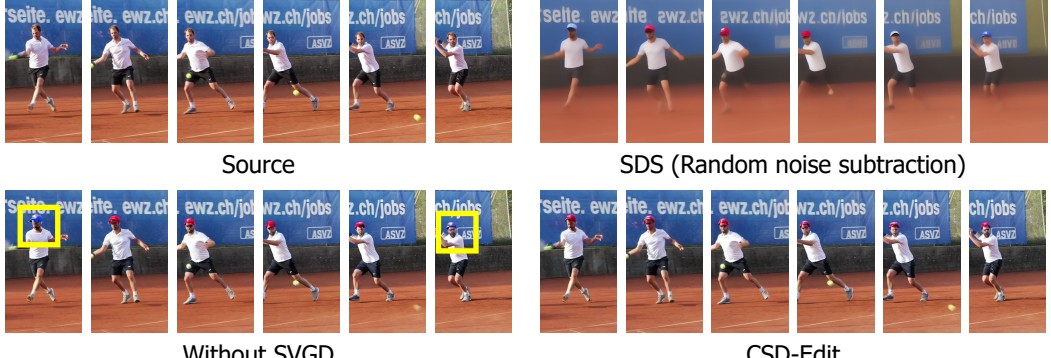

Source               SDS (Random noise subtraction)

Without SVGD              CSD-Edit

*"Give him a cap"*

Figure 6: **Ablation study**. Given a source video (top left), CSD-Edit without SVGD results in inconsistent frames (bottom left), and replacing the subtraction of image-conditioned noise in CSD-Edit to the subtraction of random noise results in loss of details and original structures (top right). CSD-Edit obtains and preserves consistency between edits without loss of semantics (bottom right).

for evaluating consistency and instruction fidelity, we also conduct a user study, given the subjective nature of editing tasks. We measure the user rankings for temporal consistency between edited frames, frame-wise instruction fidelity, and the editing quality, respectively. We use video sequences from the popular DAVIS [33] dataset at a resolution of $1920 \times 1080$. Please refer to Appendix B.2 and Appendix C.3 for a detailed description of the baseline methods and experimental setup.

Table 1 and Table 6 in Appendix C.3 summarize a quantitative comparison between CSD-Edit and the baselines. We note that CSD-Edit consistently outperforms the existing zero-shot video editing approaches in terms of both temporal consistency and fidelity to given text prompts. Furthermore, Figure 3 qualitatively demonstrates the superiority of CSD-Edit over the baselines. Specifically, CSD-Edit maintains a consistent style across all frames for both the woman and the background elements (e.g., bench, trees), ensuring a consistent degree of editing throughout the video. On the other hand, FateZero and Pix2Video result in noticeably inconsistent edits from one frame to the next. Impressively, CSD-Edit not only demonstrates temporally consistent edits compared to Gen-1, but it also excels at preserving the original semantics of the source video, even without training on a large video dataset and without requiring any architectural modifications to the diffusion model. Additional qualitative results, including video stylization and object-aware editing tasks, are in Appendix D.

### 4.3 Text-guided 3D scene editing

For the text-guided 3D scene editing experiments, we mainly compare our approach with Instuct-NeRF2NeRF (IN2N) [39]. For a fair comparison, we exactly follow the experimental setup which they used, and faithfully find the hyperparameters to reproduce their results. For evaluation, we render images at the novel views (i.e., views not seen during training), and report CLIP image similarity and LPIPS between consecutive frames in rendered videos to measure multi-view consistency, as well as CLIP image-text similarity to measure fidelity to the instruction. In addition, we conduct user studies to evaluate the multi-view consistency, instruction-fidelity, and the editing quality, respectively. Detailed explanations for each dataset sequence and training details can be found in Appendix B.3.

Figure 4, Table 2, and Table 7 in Appendix C.3 summarize the comparison between CSD-Edit and IN2N. We notice that CSD-Edit enables a wide-range control of 3D NeRF scenes, such as delicate attribute manipulation (e.g., facial expression alterations) and scene-stylization (e.g., conversion to the animation style). Especially, we notice two advantages of CSD-Edit compared to IN2N. First, CSD-Edit presents high-quality details to the edited 3D scene by providing multi-view consistent training views during NeRF optimization. In Figure 4, one can observe that CSD-Edit captures sharp details of the anime character, while IN2N results in a blurry face. Second, CSD-Edit is better at preserving the semantics of source 3D scenes, e.g., backgrounds or colors. For instance in Figure 4, we notice that CSD-Edit allows subtle changes in facial expressions without changing the color of the background or adding a beard to the face.

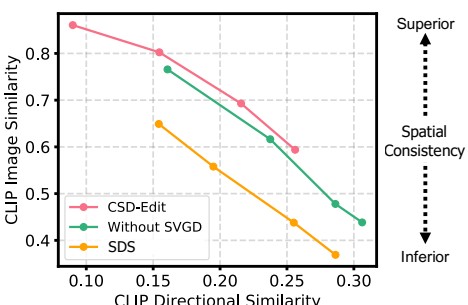

Figure 7: **Ablation on panorama image editing**. Ablation study of CSD-Edit by removing SVGD or replacing the subtraction of image-conditioned noise into the random noise (SDS).

Table 3: **Ablation on video editing**. Ablation study of CSD-Edit by removing SVGD or replacing the subtraction of the image-conditioned noise into the random noise (*i.e.,* SDS).

|  | CLIP Directional Similarity ↑ | CLIP Image Consistency ↑ | LPIPS ↓ |
|---|---|---|---|
| **CSD-Edit** | **0.320** | **0.957** | **0.236** |
| Without SVGD | 0.278 | 0.909 | 0.372 |
| SDS [26] | 0.270 | 0.884 | 0.316 |

Table 4: **Ablation on 3D scene editing**. Ablation study of CSD-Edit by removing SVGD or replacing the subtraction of the image-conditioned noise into the random noise (*i.e.,* SDS).

|  | CLIP Directional Similarity ↑ | CLIP Image Consistency ↑ | LPIPS ↓ |
|---|---|---|---|
| **CSD-Edit** | **0.239** | **0.995** | **0.043** |
| Without SVGD | 0.186 | 0.993 | 0.053 |
| SDS [26] | 0.186 | 0.994 | 0.053 |

## 4.4 Ablation study

To demonstrate the effectiveness of our method, CSD-Edit, we conduct ablation studies in each visual domain. To verify the role of Stein Variational Gradient Descent (SVGD) in ensuring consistency during the editing, we remove the kernel mixing component in Eq. (9). In addition, to verify the role of the subtracting image-conditioned noise estimate ($\epsilon_\phi^{\omega_s}$ in Eq. (9)) in obtaining high-quality visual edits, we subtract the Gaussian noise $\epsilon$ that is used to forward samples as similar to SDS [26], instead of the image-conditioned noise estimate. Again, we follow the experimental and evaluation setups in Section 4.1, Section 4.2, and Section 4.3 for the editing of each visual domain, respectively.

Figure 7, Table 3, and Table 4 show the quantitative results of the ablation studies on panorama image editing, video editing, and 3D scene editing, respectively. Note that the absence of SVGD in CSD-Edit radically alters the image, underscoring its critical role in consistency regularization. This is evidenced by lower CLIP image similarity scores, which measure the consistency between the source and edited images, and lower LPIPS scores, which evaluate the consistency between edits. In addition, the lack of mixing scores between samples leads to abrupt and undesirable changes in the source, as shown in Figure 8. Further, replacing image-conditioned noise subtraction with random noise subtraction results in a loss of the original structure of the source image, which significantly degrades CLIP image similarity and CLIP directional similarity.

In Figure 6, we provide a qualitative validation of the effectiveness of our method on video editing. CSD-Edit consistently edits a source video by adding a red cap to a man's head when given the instruction "Give him a cap." However, with SVGD, the edits between frames are inconsistent, e.g. both blue and red caps appear on the edited frames. In addition, if we set the baseline noise as the random noise injected into the source and target images, each frame becomes blurred and loses the original structures, such as blurred legs and backgrounds. For more qualitative results of our ablation studies, please refer to Figure 13 in Appendix D.

## 5 Related work

Following the remarkable success of text-to-image diffusion models [4, 20, 1, 2, 43], numerous works have attempted to exploit rich knowledge of them for various visual editing tasks including images [10, 44, 13, 45, 14, 12, 15], videos [46, 25], 3D scenes [39], etc. However, extending existing image editing approaches to more complex visual modalities often faces a new challenge; consistency between edits, e.g., spatial consistency in high-resolution images, temporal consistency in videos, and multi-view consistency in 3D scenes. While prior works primarily focus on designing task-specific methods [24, 22, 25] or model fine-tuning for complex modalities [46], we present a modality-agnostic novel method for editing, effectively capturing consistency between samples.

The most related to our work is DreamFusion [26], which introduced Score Distillation Sampling (SDS) for the creation of 3D assets, leveraging the power of text-to-image diffusion models. Despite the flexible merit of SDS to enable the optimization of arbitrary differentiable operators, most

*"Turn sheeps into tigers"*

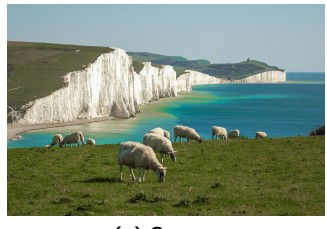 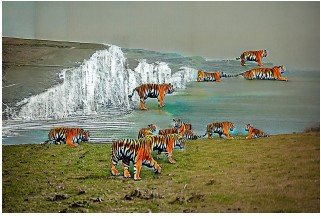 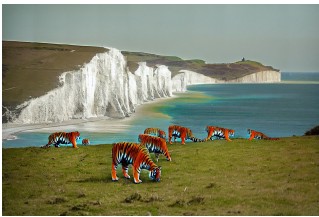

|  (a) Source  |  (b) CSD-Edit without SVGD  |  (c) CSD-Edit  |

Figure 8: **Ablation study for the effect of SVGD**. Our kernel mixing scores in CSD-Edit act as a regularizer that prevents abrupt changes in images, ensuring better consistency: the tiger is generated in the unwanted region of a source image when SVGD is not applied in the update.

works mainly focus on applying SDS to enhance the synthesis quality of 3D scenes by introducing 3D-specific frameworks [47, 48, 49, 50, 51]. Although there exists some work to apply SDS for visual domains other than 3D assets, they have limited their scope to image editing [52], or image generation [53]. Here, we argue that the current main challenge preventing the wider application of SDS, especially in higher-dimensional visual manipulations beyond single 2D images at fixed resolutions, is the lack of control over inter-sample consistency. To the best of our knowledge, our work is the first to identify this challenge and to lay the novel foundations for principled adaptation of text-to-image diffusion models to more diverse and high-dimensional visual manipulations.

## 6    Conclusion

In this paper, we propose Collaborative Score Distillation (CSD) for consistent visual synthesis and manipulation. CSD is built upon Stein variational gradient descent, where multiple samples share their knowledge distilled from text-to-image diffusion models during the update. Furthermore, we propose CSD-Edit that gives us consistent editing of images by distilling minimal, yet sufficient information from instruction-guided diffusion models. We demonstrate the effectiveness of our method in text-guided translation of diverse visual contents, such as in high-resolution images, videos, and real 3D scenes, outperforming previous methods both quantitatively and qualitatively.

**Limitations and future works.**   Since we use pre-trained text-to-image diffusion models, obtained results are often imperfect due to the inherent inability of diffusion models to understand language. Furthermore, our method relies on generative priors derived from large text-to-image diffusion models, which may inadvertently contain biases due to the auto-filtering process applied to the vast training dataset. However, we believe that employing Consistent Score Distillation (CSD) can assist us in identifying and understanding such undesirable biases. By leveraging the inter-sample relationships and aiming for consistent generation and manipulation of visual content, our method provides a valuable avenue for comprehending the interaction between samples and prompts. Further exploration of this aspect represents an intriguing future direction.

In addition, although our primary interest is in the editing (not the generation) of panoramic images, videos, or 3D scenes, we believe that CSD has the potential to be used in their generation. As presented in Section C.2 in Appendix C, we show how CSD can improve generation performance over SDS [26] on text-to-3D generation experiments. In particular, we verify the effect of CSD in improving the geometry and quality of text-to-3D generation. In this sense, exploring this aspect of synthesis with CSD could be an interesting research topic and we leave it for future work.

**Societal impact.**   Our research introduces a comprehensive image editing framework that encompasses various modalities, including high-resolution images, videos, and 3D scenes. While it is important to acknowledge that our framework might be potentially misused to create fake content, this concern is inherent to image editing techniques as a whole. We expect future research on the detection of generated visual content.

## Acknowledgement

This work was supported by Institute of Information & communications Technology Planning & Evaluation (IITP) grant funded by the Korea government(MSIT) (No.2019-0-00075, Artificial Intelligence Graduate School Program(KAIST); No.2021-0-02068, Artificial Intelligence Innovation Hub; No.2022-0-00184, Development and Study of AI Technologies to Inexpensively Conform to Evolving Policy on Ethics). This work is in partly supported by Google Research grant and Google Cloud Research Credits program. We thank Jason Baldridge, Yi-Hsuan Tsai, Sihyun Yu, and Sangwoo Mo for reviewing our manuscript, and anonymous reviewers for their helpful comments and suggestions.

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

# Appendix

Website: https://subin-kim-cv.github.io/CSD

## A   Technical details

In this section, we provide detailed explanations on the proposed methods, CSD and CSD-Edit.

**CSD derivation.**   Consider a set of parameters $\{\theta_i\}_{i=1}^N$ which generates images $\mathbf{x}^{(i)} = g(\theta_i)$. For each timestep $t \sim \mathcal{U}(t_{\texttt{min}}, t_{\texttt{max}})$, we aim at minimizing the following KL divergence

$$D_{\texttt{KL}}\big(q(\mathbf{x}_t^{(i)}|\mathbf{x}^{(i)} = g(\theta_i))\|p_\phi(\mathbf{x}_t; y, t)\big)$$

for each $i = 1, 2, \ldots, N$ via SVGD using Eq. (7). To this end, we approximate the score function, (i.e., gradient of log-density) by the noise predictor from diffusion model as follows:

$$\nabla_{\mathbf{x}_t^{(i)}} \log p_\phi(\mathbf{x}_t^{(i)}; y, t) \approx -\frac{\boldsymbol{\epsilon}_\phi(\mathbf{x}_t^{(i)}; y, t)}{\sigma_t}.$$

Then, the gradient of score function with respect to parameter $\theta_i$ is given by

$$\nabla_{\theta_i} \log p_\phi(\mathbf{x}_t^{(i)}; y, t) = \nabla_{\mathbf{x}_t^{(i)}} \log p_\phi(\mathbf{x}_t^{(i)}; y, t)\frac{\partial \mathbf{x}_t^{(i)}}{\partial \theta_i} \approx -\frac{\alpha_t}{\sigma_t}\boldsymbol{\epsilon}_\phi(\mathbf{x}_t^{(i)}; y, t)\frac{\partial \mathbf{x}^{(i)}}{\partial \theta}, \qquad (10)$$

for each $i = 1, \ldots N$. Finally, to derive CSD, we plug Eq. (10) to Eq. (7) to attain Eq. (8). Also, we subtract the noise $\boldsymbol{\epsilon}$, which helps reduce the variance of the gradient for better optimization. Following DreamFusion [26], we do not compute the Jacobian of U-Net. At a high level, CSD takes the gradient update on each $\mathbf{x}^{(i)}$ using SVGD and updates $\theta_i$ by simple chain rule without computing the Jacobian. This formulation makes CSD a straightforward generalization to SDS for multiple samples and leads to an effective gradient for optimizing consistency among batches of samples.

**CSD-Edit derivation.**   As mentioned above, we subtract the random noise to reduce the variance of CSD gradient estimation. This is in a similar manner to the variance reduction in policy gradient [54], where having a proper baseline function results in faster and more stable optimization. Using this analogy, our intuition is built upon that setting a better baseline function can ameliorate the optimization of CSD. However, as shown in Figures 6 and Figure 13 in our manuscript, the default choice of SDS (random noise) results in highly blurred outputs, because the noise-denoising process of SDS blurs the image. Therefore, we propose to use image-conditional noise estimation as a baseline function in image editing via CSD-Edit, so that the diffusion noise only alters the part where the text instructs to change, by allowing CSD-Edit to optimize the latent driven only by the influence of the instruction prompts. This approach is supported by the principle of Wasserstein gradient flow [55], where the optimal gradient flow in variational inference is given by the difference between the target score function and the source score function.

Moreover, we notice that similar observations were proposed in Delta Denoising Score (DDS) [52], where they introduced an image-to-image translation method that is based on SDS, and the difference of the noise estimate from target prompt and that from source prompt are used. Our CSD can be combined with DDS by changing the noise difference term as follows:

$$\Delta \boldsymbol{\mathcal{E}}_t = \boldsymbol{\epsilon}_\phi(\mathbf{x}_t; y_{\texttt{tgt}}, t) - \boldsymbol{\epsilon}_\phi(\tilde{\mathbf{x}}_t; y_{\texttt{src}}, t),$$

where $\mathbf{x}$ and $\tilde{\mathbf{x}}$ are target and source images, $y_{\texttt{tgt}}$ and $y_{\texttt{src}}$ are target and source prompts. However, we found that CSD-Edit with InstructPix2Pix is more amenable in editing real images as it does not require a source prompt. Finally, we remark that CSD-Edit can be applied to various text-to-image diffusion models such as ControlNet [17], which we leave for future work.

# B Implementation details

**Setup.** For the experiments with CSD-Edit, we use the publicly available pre-trained model of Instruct-Pix2Pix [14][2] by default. We perform CSD-Edit optimization on the output space of Stable Diffusion [4] autoencoder. We use SGD optimizer with step learning rate decay, without adding weight decay. We set $t_{\min} = 0.2$ and $t_{\max} = 0.5$, where original SDS optimization for DreamFusion used $t_{\min} = 0.2$ and $t_{\max} = 0.98$. This is because we do not generally require a large scale of noise in editing. We use the guidance scale $\omega_y \in [3.0, 15.0]$ and image guidance scale $\omega_s \in [1.5, 5.0]$. We find that our approach is less sensitive to the choice of image guidance scale, yet a smaller image guidance scale is more sensitive to editing. All experiments are conducted on AMD EPYC 7V13 64-Core Processor and a single NVIDIA A100 80GB. Throughout the experiments, we use OpenCLIP [56] `ViT-bigG-14` model for evaluation.

## B.1 Panorama image editing

To edit a panorama image, we first encode into the Stable Diffusion latent space (i.e., downscale by 8), then use a stride size of 16 to obtain multiple patches. Then we select a $B$ batch of patches to perform CSD-Edit. Note that we perform CSD-Edit and then normalize by the number of appearances as mentioned in Section 3.3. Note that our approach performs well even without using small batch size, e.g., for an image of resolution 1920×512, there are 12 patches and we use $B = 4$.

For experiments, we collect 32 panorama images and conduct 5 artistic stylizations: "turn into Van Gogh style painting", "turn into Pablo Picasso style painting", "turn into Andy Warhol style painting", "turn into oriental style painting", and "turn into Salvador Dali style painting". We use learning rate of 2.0 and image guidance scale of 1.5, and vary the guidance scale from 3.0 to 10.0.

## B.2 Video editing

We edit video sequences in DAVIS 2017 [33] by sampling 24 frames at the resolution of 1920×1080 from each sequence. Then, we resize all frames into 512×512 resolution and encode all frames each using Stable Diffusion. We use learning rate $[0.25, 2]$ and optimize them for $[200, 500]$ iterations.

## B.3 3D scene editing

Following Instruct-NeRF2NeRF [39], we first pretrain NeRF using the *nerfacto* model from NeRFStudio [57], training it for 30,000 steps. Next, we re-initialize the optimizer and finetune the pre-trained NeRF model with edited train views. In contrast to Instruct-NeRF2NeRF, which edits one train view with Instruct-Pix2Pix after every 10 steps of update, we edit a batch of train views (batch size of 16) with CSD-Edit after every 2000 steps of update. The batch is randomly selected among the train views without replacement.

---

[2] https://github.com/timothybrooks/instruct-pix2pix

## C  More experimental results

### C.1  Compositional editing

Recent works have shown the ability of text-to-image diffusion models in compositional *generation* of images handling multiple prompts [58, 59]. Here, we show that CSD-Edit can extend this ability to compositional *editing*, even at panorama-scale images which require a particular ability to maintain far-range consistency. Specifically, we demonstrate that one can edit a panorama image to follow different prompts on different regions while keeping the overall context uncorrupted.

Given multiple textual prompts $\{y_k\}_{k=1}^{K}$, the compositional noise estimate is given by

$$\boldsymbol{\epsilon}_\phi(\mathbf{x}_t; \{y_k\}_{k=1}^{K}, t) = \sum_{k=1}^{K} \alpha_k \boldsymbol{\epsilon}_\phi^\omega(\mathbf{x}_t; y_k, t),$$

where $\alpha_k$ are hyperparameters that regularize the effect of each prompt. When applying compositional generation to the panorama image editing, the challenge lies in obtaining image that is smooth and natural within the region where the different prompts are applied. To that end, for each patch of an image, we set $\alpha_k$ to be the area of the overlapping region between the patch and region where prompt $y_k$ is applied. Also, we normalize to assure $\sum_k \alpha_k = 1$. In Figure 10, we illustrate some examples on compositional editing of a panorama image. For instance, given an image, one can change into different weathers, different seasons, or different painting styles without leaving artifacts that hinder the spatial consistency of an image.

### C.2  Text-to-3D generation with CSD

We explore the effectiveness of CSD in text-to-3D generation tasks following DreamFusion [26]. We train a coordinate MLP-based NeRF architecture from scratch using text-to-image diffusion models. Since the pixel-space diffusion model that DreamFusion used [26] is not publicly available, we used an open-source implementation of pixel-space text-to-image diffusion model.[3] Given a set of text prompts, we run both DreamFusion and DreamFusion with CSD with a fixed seed. Our experiments in this section are based on Stable-DreamFusion [60], a public re-implementation of DreamFusion, given that currently the official implementation of DreamFusion is not available on public.

**Setup.**  We use vanilla MLP based NeRF architecture [27] with 5 ResNet [61] blocks. Other regularizers such as shading, camera and light sampling are set as default in [60]. We use view-dependent prompting given the sampled azimuth angle and interpolate by the text embeddings. We use Adan [62] optimizer with learning rate warmup over 2000 steps from $10^{-9}$ to $2 \times 10^{-3}$ followed by cosine decay down to $10^{-6}$. We use batch size of 4 and optimize for 10000 steps in total, where most of the case sufficiently converged at 7000 to 8000 steps. For the base text-to-image diffusion model, we adopt `DeepFloyd-IF-XL-v1.0` since we found it way better than the default choice of Stable Diffusion in a qualitative manner. While the original DreamFusion [26] used guidance scale of 100 for their experiments, we find that guidance scale of 20 works well for `DeepFloyd`. We selected 30 prompts used in DreamFusion gallery[4] and compare their generation results via DreamFusion from the standard SDS and those from our proposed CSD. We use one A100 (80GB) GPU for each experiment, and it takes ∼5 hours to conduct one experiment. For CSD implementation, we use LPIPS [42] as a distance of RBF kernel. Note that LPIPS gives more computational cost than the usual $\ell_2$-norm based RBF kernel. The LPIPS is computed between two rendered views of size 64×64. For the kernel bandwidth, we use $h = \frac{\text{med}^2}{\log B}$, where med is a median of the pairwise LPIPS distance between the views, $B$ is the batch size.

For evaluation, we render the scene at the elevation at 30 degree and capture at every 30 degree of azimuth angle. Then we compute the CLIP image-text similarity between the rendered views and input prompts. We measure similarities for both textured views (RGB) and textureless depth views (Depth). We also report Frechet Inception Distance (FID) between the RGB images and ImageNet validation dataset to evaluate the quality and diversity of rendered images compared to natural images.

---

[3]https://github.com/deep-floyd/IF
[4]https://dreamfusion3d.github.io/gallery.html

Table 5: **Text-to-3D**. Quantitative comparison between CSD and SDS under on text-to-3D generation via DreamFusion [26]

| | CLIP Similarity Color ↑ | CLIP Similarity Geo ↑ | FID ↓ |
|---|---|---|---|
| SDS [26] | 0.437 | 0.322 | 259.4 |
| **CSD (Ours)** | **0.447** | **0.345** | **247.1** |

**Results.**   In Table 5, we report the evaluation results of CSD on text-to-3D generation comparison to DreamFusion. Remark that CSD presents better CLIP image-text similarities in both RGB and Depth views. Also, CSD achieves lower FID score showing its better quality on generated samples. Since we used the same random seed in generating both CSD and DreamFusion, the shapes and colors are similar. However, the results show that CSD obtains finer details in its generations.

In Figure 14, we qualitatively compare the baseline DreamFusion (SDS) and ours. We empirically observe three advantages of using CSD over SDS. First, CSD provides better quality compared to SDS. SDS often suffers from Janus problem, where multiple faces appear in a 3D object. We found that CSD often resolves Janus problem by showing consistent information during training. See the first row of Figure 14. Second, CSD can give us better fine-detailed quality. The inconsistent score distillation often gives us blurry artifact or undesirable features left in the 3D object. CSD can handle this problem and results in higher-quality generation, e.g., Figure 14 second row. Lastly, CSD can be used for improving diversity. One problem of DreamFusion, as acclaimed by the authors, is that it lacks sample diversity. Thus, it often relies on changing random seeds, but it largely alters the output. On the other hand, we show that CSD can obtain alternative sample with only small details changed, e.g., Figure 14 third row. Even when SDS is successful, CSD can be used in generating diverse sample.

## C.3   User study

Given the subjective nature of editing tasks, we additionally conduct subjective user studies, where we ask three questions in evaluating the editing quality of both CSD-Edit and baselines: the consistency of the edited results, frame-wise image instruction fidelity, and the editing quality. For each of the three studies, we asked 20 subjects to rank different methods. As shown in Table 6 and Table 7, our method, CSD-Edit, consistently outperforms other baselines achieving the best user preferences across all three aspects.

Table 6: **User study on video editing**.

| | Temporal Consistency ↓ | Instruction Fidelity ↓ | Editing Quality ↓ |
|---|---|---|---|
| FateZero [22] | 2.37 | 2.05 | 2.12 |
| Pix2Video [25] | 2.36 | 2.35 | 2.28 |
| **CSD-Edit (Ours)** | **1.27** | **1.6** | **1.6** |

Table 7: **User study on 3D scene editing**.

| | Temporal Consistency ↓ | Instruction Fidelity ↓ | Editing Quality ↓ |
|---|---|---|---|
| IN2N [14] | 1.61 | 1.69 | 1.71 |
| **CSD-Edit (Ours)** | **1.39** | **1.31** | **1.29** |

## C.4 Computation time comparison

Regarding the computational efficiency of our method, we measure the computation times and compare them with the baselines of each task. All these evaluations were conducted on a single NVIDIA A100 80GB and AMD EPYC 7V13 64-Core Processor.

For panorama image editing experiments, we compare our method with the baseline where applying Instruct-Pix2Pix to patches cropped with overlapping (InstructPix2Pix + Overlapping Patches). Note that the computation time of both methods depends on the input image resolution, whereas we show that our method becomes more efficient as the resolution goes higher. Table 8 shows how the computation time (total time in seconds) differs by the size of the input image. Here, the baseline method requires computing noise estimates of every patch at each diffusion step, while our method only requires computing a minibatch of patches per iteration.

Table 8: **Computation time comparison on panorama image editing**.

| Method | Resolution | Total time (sec.) |
|---|---|---|
| InstructPix2Pix + Overlapping Patches | 1920×640 | **62** |
| **CSD-Edit (Ours)** | 1920×640 | 68 |
| InstructPix2Pix + Overlapping Patches | 3968×4352 | 487 |
| **CSD-Edit (Ours)** | 3968×4352 | **275** |

In video editing experiments, we measure the total computation time of our method and baseline methods. As shown in Table 9, our method, CSD-Edit, consistently outperforms the baselines, even taking the shortest time in editing (×3.3 times faster than FateZero [22], ×1.3 times faster than Pix2Video [25]. Note that the computational efficiency of our method, CSD-Edit, on video editing can be easily improved by reducing the number of iterations for optimization while using a larger learning rate. Here, using a higher learning rate changes frames more radically, thus slightly degrading the frame consistency, however, the effect is meager as shown in CLIP Image consistency and LPIPS metrics.

Table 9: **Computation time comparison on video editing**.

| | Total time (sec). | CLIP Directional Similarity ↑ | CLIP Image Consistency ↑ | LPIPS ↓ |
|---|---|---|---|---|
| FateZero [22] | 192 | $0.312_{\pm 0.003}$ | $0.948_{\pm 0.001}$ | $0.264_{\pm 0.002}$ |
| Pix2Video [25] | 77 | $0.229_{\pm 0.001}$ | $0.948_{\pm 0.001}$ | $0.282_{\pm 0.001}$ |
| **CSD-Edit (Ours)** | **59** | $0.320_{\pm 0.001}$ | $0.955_{\pm 0.001}$ | $0.243_{\pm 0.001}$ |
| **CSD-Edit (Ours)** | 423 | $\mathbf{0.319_{\pm 0.002}}$ | $\mathbf{0.957_{\pm 0.001}}$ | $\mathbf{0.235_{\pm 0.001}}$ |

In text-to-3D generation experiments, when directly comparing with DreamFusion [26](or SJC [63]), there is a slight increase in the computational cost due to the usage of LPIPS metric as distance for RBF kernel. For instance, it takes 1 hour to generate a 3D model with DreamFusion, while using CSD takes 84 minutes. We note that the increment is not substantial, however, ours yields better qualities with finer details, as shown in Table 5 and Figure 14 in the Appendix.

## C.5 More analysis about the batch size

We denote N as the total number of images (e.g., total patches of a panorama image or total frames of a video) and we update B minibatch of samples per iteration. Intuitively, using a large B would encourage more consistency among samples, but it is more computationally expensive. Also, we observed that using large batch sizes dilutes the effect of editing. Thus, the batch size controls the trade-off between computation time, editing quality, and preserving consistency. To further verify our choice of B, we provide additional ablation studies on the panorama image editing experiments. Given the same experimental setup as in Section 4.1 with fixed guidance scale=7.5, we swept over B = 4, 8, 12, and measured the CLIP source-target image similarity, CLIP directional image-text similarity, and computation time (iteration per second for total 200 iterations, measured on single A100 40GB GPU). Table 10 demonstrates the effect of batch size.

Table 10: **Ablation study about the batch size**.

| Batch size | Total time (iter / sec.) | CLIP Directional Similarity ↑ | CLIP Image Similarity ↑ |
|---|---|---|---|
| B=4 | 2.86 | 0.639 | 0.240 |
| B=8 | 1.47 | 0.692 | 0.217 |
| B=12 | 1.02 | 0.739 | 0.195 |

"*Turn penguins into chickens*"

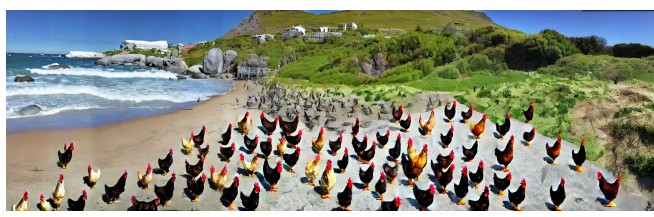

(a) minibatch B=3

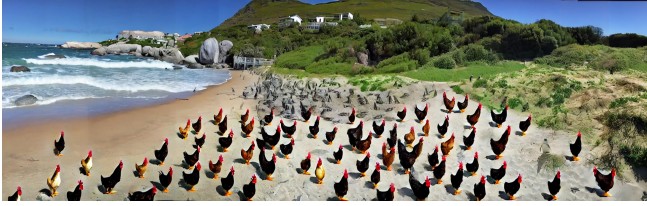

(b) minibatch B=12

Figure 9: **Ablation study: the effect of batch size**. One can control the diversity of generated output, e.g., the same penguins are changed to more diverse chickens, by choosing an appropriate batch size.

# D  Additional qualitative results

Source

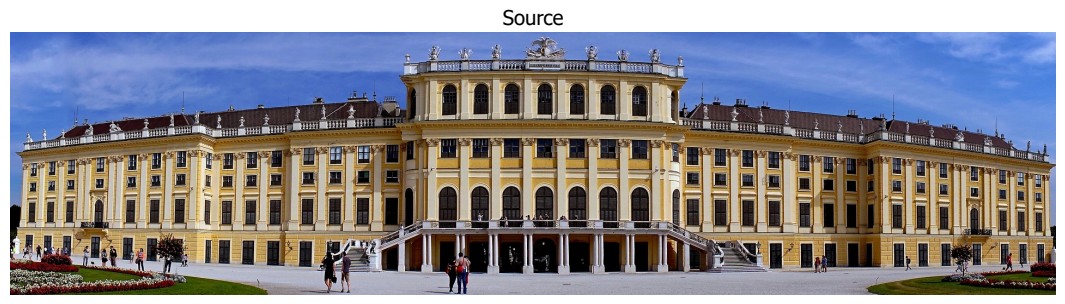

"Turn into sunny weather"  "Turn into rainy weather"  "Turn into snowy weather"

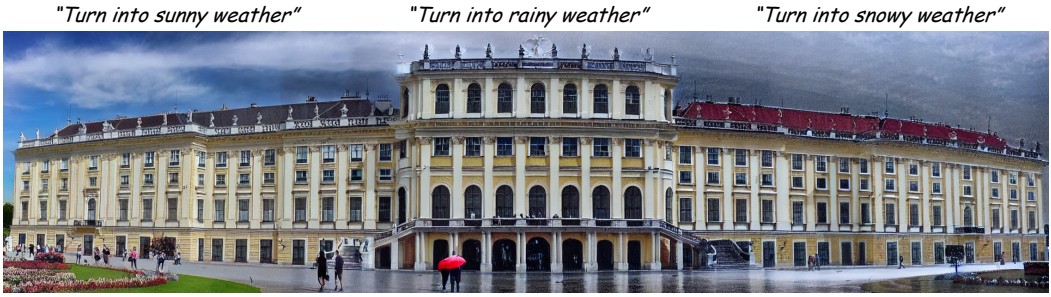

"Turn into spring"  "Turn into fall"

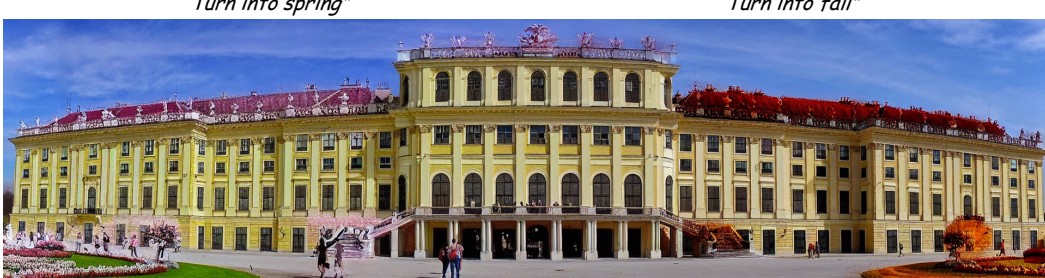

"Turn into Van Gogh style painting"  "Turn into Paul Gauguin style painting"

Figure 10: **Compositional image editing**. CSD-Edit demonstrates the ability to edit consistently and coherently across patches in panorama images. This provides the unique capability to manipulate each patch according to different instructions while maintaining the overall structure of the source image. Remarkably, CSD-Edit ensures a smooth transition between patches, even when different instructions are applied.

Source

"Turn sheeps into wolves"

"Turn sheeps into kangaroos"

"Turn sheeps into polar bears"

"Turn sheeps into reindeers"

Source

"Turn penguins into chickens"

"Turn penguins into bears"

"Turn penguins into pandas"

"Turn penguins into sea lions"

Figure 11: **Object editing**. CSD-Edit can edit many objects in a wide panorama image consistently in accordance with the given instruction while preserving the overall structure of source images.

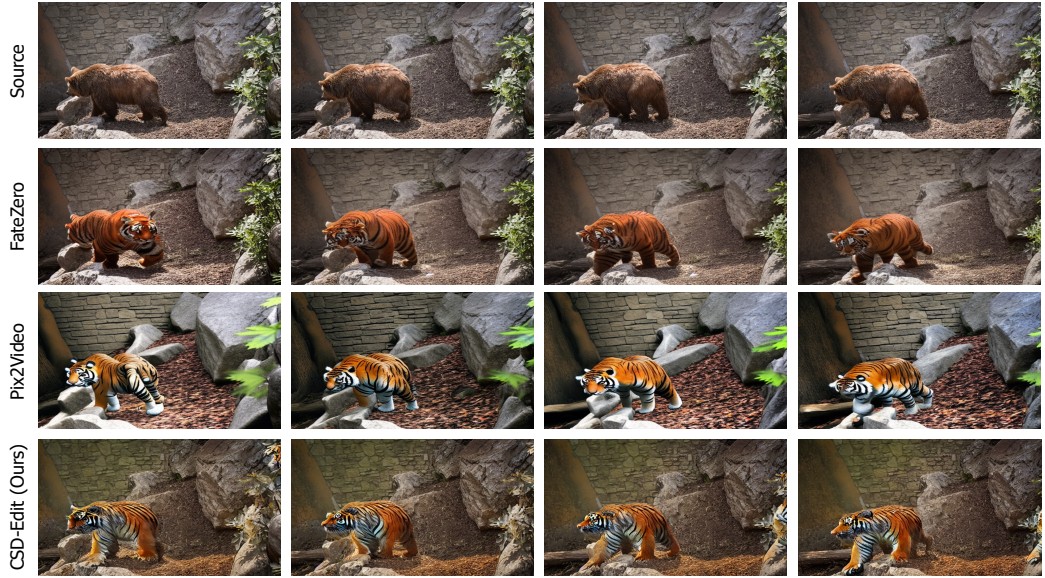

"Turn a bear into a tiger"

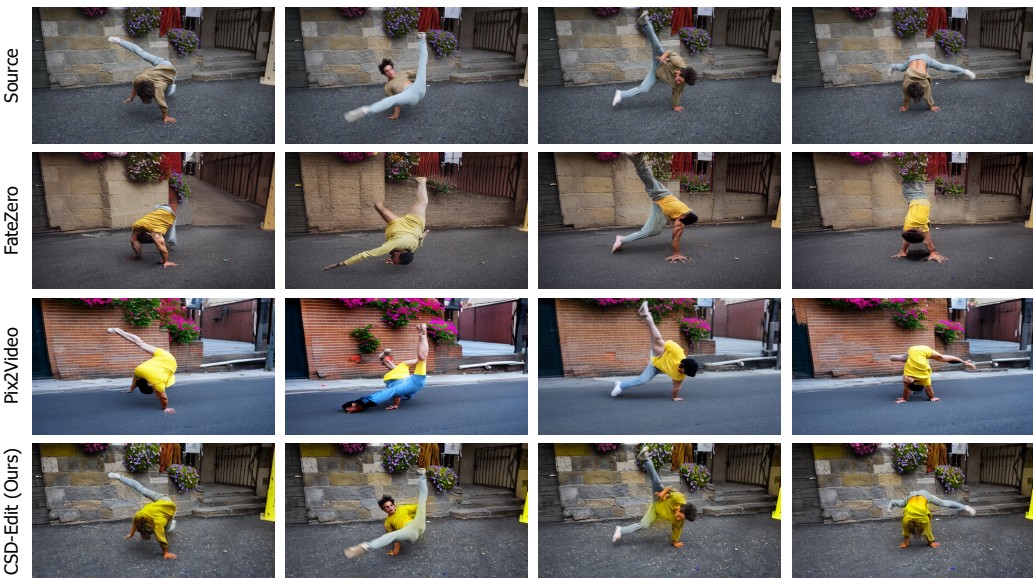

"Give him a yellow T-shirt"

Figure 12: **Video editing**. CSD-Edit demonstrates various editing from an object (e.g., tiger) to attributes (e.g., color) while providing consistent edits across frames and maintaining the overall structure of a source video.

Source

*"Turn into Van Gogh style painting"*

*"Turn into Pablo Picasso style painting"*

*"Turn into Andy Warhol style painting"*

*"Turn into oriental style painting"*

*"Turn into Salvador Dali style painting"*

|  CSD-Edit  |  CSD-Edit without SVGD  |  CSD-Edit with Random Noise  |

Figure 13: **Ablation study: SVGD and random noise**. As illustrated, edits across different patches are not consistent without SVGD. Also, when using random noise as baseline noise, it loses the content and the detail of the source image.

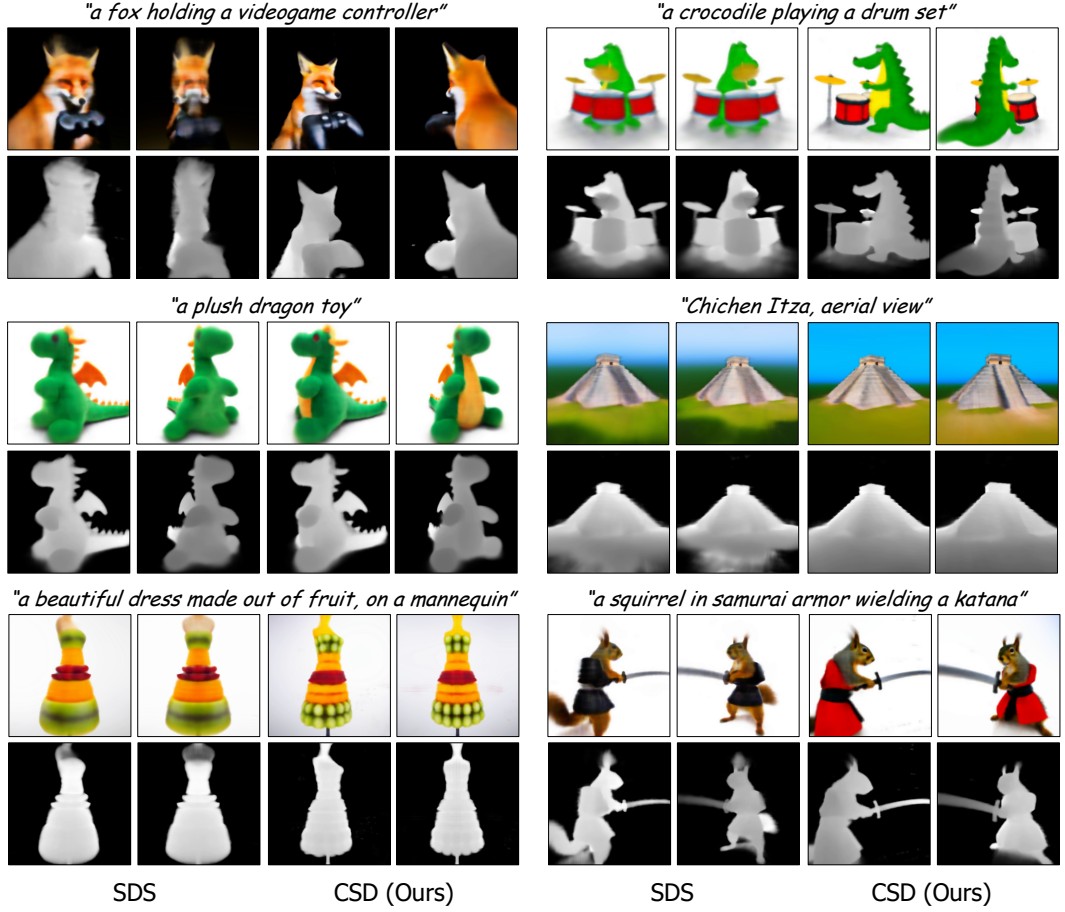

Figure 14: **Text-to-3D generation examples**. (First row) CSD helps to capture coherent geometry compared to using SDS. (Second row) CSD allows learning finer details than SDS. (Third row) CSD can provide diverse and high-quality samples without changing random seeds.

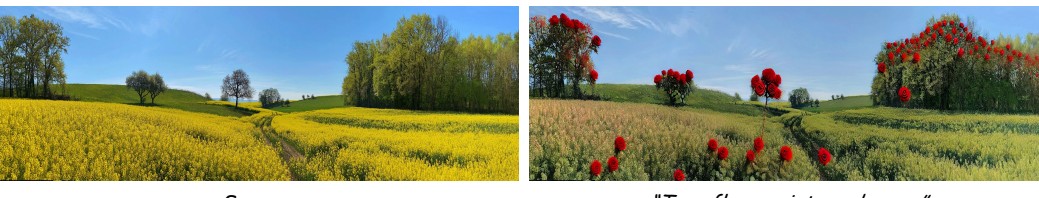

Source                *"Turn flowers into red roses"*

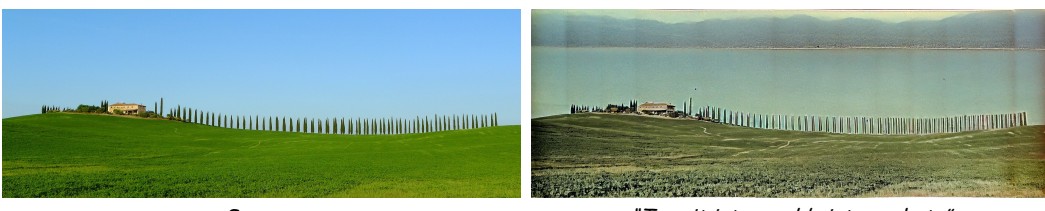

Source                *"Turn it into an old vintage photo"*

Figure 15: **Limitations**. (First row) CSD-Edit often manipulates undesirable contents due to the inherent inability of Instruct-Pix2Pix model. (Second row) CSD-Edit often produces artifacts on the image due to the patch-wise update.

## E    Limitations

As our method leverages pre-trained Instruct-Pix2Pix, it inherits the limitations of it such as undesirable changes to the image due to the biases. Also, as described in [14], Instruct-Pix2Pix is often unable to change viewpoints, isolate a specific object, or reorganize objects within the image.

When editing a high-resolution image by dividing it into patches, it often remains an artifact at the edge of the patches, especially at the corner side of an image. This is due to that the patches at the corner are less likely to be sampled during the optimization. See Figure 15 for examples.

When editing a video, the edited video often shows a flickering effect due to the inability of the Stable Diffusion autoencoder to compress the video. We believe that using CSD-Edit with video diffusion models trained on video datasets can possibly overcome this problem.

