# OpenReview forum: "Collaborative Score Distillation for Consistent Visual Editing"
_NeurIPS.cc/2023/Conference — NeurIPS 2023 poster_

### Official Review · Reviewer_yGH2 · 2023-06-22

**Soundness:** 2 fair
**Presentation:** 2 fair
**Contribution:** 2 fair
**Rating:** 4
**Confidence:** 4

**Summary:**

The paper introduces an approach to achieve consistent visual editing by leveraging a pre-trained pix2pix diffusion model. The authors propose a generalization of the SDS loss (originating from DreamFusion) to a CSD loss, which utilizes Stein variational gradient descent. This new loss function enables the joint distillation of multiple samples from a text-to-image diffusion model. The CSD loss is applicable to diverse visual editing scenarios, including panorama images, videos, and 3D scenes. Additionally, the CSD loss can be utilized for text-to-3D generation as well.

**Strengths:**

- The authors provided results for various visual editing applications, such as panorama images, videos, and 3D scenes, as well as text-to-3D generation. This demonstrates the promise of the suggested method's flexibility for numerous applications.
- The paper reads well and flows smoothly.
- The authors presented all the required preliminaries, making the paper self-contained for newcomers.

**Weaknesses:**

- The paper lacks reproducibility due to several crucial missing details:
  - The paper does not specify the meaning or selection process of parameter N in all applications, nor how to choose the final parameter theta from the set of N parameters.
  - The formulation is missing information on the aggregation over multiple views/frames/crops, making it unclear how to implement this aspect.
  - The explicit specifications of classifier-free guidance weights are missing, particularly in the 3D generation application where DreamFusion originally requires a high weight to produce desired results.
- I don't quite undertsand why the additional epsilon is  required in both equations 8 and 9, where from equation 7 it is clear that only the score (epsilon_phi) is given. I find this to be very not justified, nor the transformation from random noise to the image-conditioned prediction in equation 9. It seems like there were several options to take here instead as well, and the specific choice seems arbitrary. The provided ablation in section 4.4 is insufficient to justify these choices, as it does not cover all possible alternatives (such as using only the score or other deterministic predicted noise).
- The paper lacks preliminary information on multidiffusion, despite being depicted in figure 2. The related work section is generally lacking and should elaborate on other methods beyond DreamFusion.
- Since the suggested method is presented as a generalization of the SDS loss, a more extensive comparison to SDS should have been provided in all applications. This is an important baseline that is currently missing.
- Figure 1 is overcrowded and confusing. It would be clearer if the generated panorama on the left side corresponded to the text description rather than being identical to the source image.

**Questions:**

- What is the reason behind the focus on editing applications in most of the experiments? Was panorama generation also explored by the authors?
- How does the paper provide an explanation for the observed comparison in the panorama editing results shown in figure 2? Was there an investigation into the potential use of higher weighting parameters for the baselines or other configurations?
- The results obtained for text-to-3D generation appear to be quite similar to the results achieved with DreamFusion using the SDS loss. Can the authors offer an explanation for this similarity in performance?

**Limitations:**

The authors discussed limitation, however the limitation figure is shown soley in the appendix. Morover, the authors did not discussed training times which could serves as on of the limitations.

---

> ### Author Rebuttal · Authors · 2023-08-09
>
> Dear reviewer yGH2,
>
> We sincerely appreciate your efforts and comments to improve the manuscript. We respond to your comment in what follows.
>
> ---
>
> **[W1]  Lack of detailed explanation for reproducibility**
>
> As for reproducibility, we comprehensively included the details required to implement our method, both in the main manuscript and appendix:
> To clarify, N denotes the total number of images and we select B minibatch of samples at each iteration to update. The implementation details are in Appendix D. Also, we provide additional ablation studies on the effect of batch size B in response [Q3] of Reviewer qUzm.
> Implementation details regarding aggregation can be found in Section 3.3 and Appendix D.
> Details on classifier-free guidance weights can be found in Appendix D for text-guided editing tasks, and in Appendix B.2 for text-to-3D generation
>
> For further clarification, we offer line-by-line explanations alongside our submitted code for video editing. The submitted code shows that B is determined by the number of video frames, i.e.,B = N (line 132), how the final edited images are obtained (line 247), and where the combination of scores from multiple samples for gradient updates occurs (lines 227 and 229).
>
>
> Lastly, we note that the code has been submitted as part of the supplementary material. We advocate for reproducible research and will open-source our code that reproduces results in the paper.
>
> ---
>
> **[W2]  Detailed analysis about the choice of subtracted baseline noise**
>
> Our rationale behind the image-conditional noise is elaborated in Appendix A, as well as in Sections 3.2 and Section 4.4. Furthermore, we elucidate the choice of using image-conditioned noise as follows:
>
> - Subtraction of random noise: Directly applying Eq. (7) solely with the score function of the target distribution leads to a high variance of gradient, which can severely hinder convergence. Thus, as also demonstrated in DreamFusion, subtracting random noise is crucial as an effective regularization.
> - Introduction of image-conditional noise: However, as shown in Figures 7 and 12 in our manuscript, the default choice of SDS results in severely blurred outputs, as the noise-denoise process of SDS blurs the image. Thus, we propose to subtract image-conditional noise so that the diffusion noise only alters the part where the text instructs to change. This approach is supported by the principle of Wasserstein gradient flow, where the optimal gradient flow in variational inference is given as the difference between the target score function and source score function (See response [Q1] to Reviewer v8pz for further details).
> - Other choices for baseline noise: Recent works show different choices in subtracting baseline noise: Delta Denoising Score [1] estimates the noise of the source image by providing a suitable source prompt, and ProlificDreamer [2] fine-tunes U-Net to obtain the noise of the source distribution (Also, refer to the response [Q1] to Reviewer v8pz). However, those are not favorable in our cases as source prompts are not given in real image-editing nor fine-tuning U-Net is computationally expensive.
>
> ---
>
> **[W3]  Missing information about detailed explanation of baseline methods**
>
> In response, we will elaborate more details about baseline methods and clarify the difference from our method in our final manuscript for a comprehensive understanding.
>
> ---
> **[W4] Lack of extensive comparison with SDS in visual editing experiments**
>
> While SDS is the most relevant approach to our method, the quality of SDS edited images is superseded by more recent works, which we considered as baselines for comparison, as explained in [W2]. Thus, we primarily compared our method with much stronger, state-of-the-art baselines of each modality to show its remarkable performance. Nevertheless, we compared our method with SDS in ablation studies, as shown in Figure 7, and Appendix C (random noise stands in for SDS), and qualitative examples are in Figure 12. We will clarify this more explicitly in our final manuscript.
>
> ---
> **[W5]  Concept figure is confusing**
>
> Thank you for the detailed suggestions. We revised Figure 1 in supplementary PDF to better illustrate our method in overview.
>
> ---
> **[Q1]  Reasons behind the focus on editing applications rather than generation**
>
> We primarily focused on editing applications in most of our experiments due to the inherent limitations associated with SDS. Specifically, SDS tends to converge towards certain modes, resulting in blurred outputs with fewer details. Given this issue, we have opted to focus on text-guided editing of panorama images utilizing image-conditioned noise, rather than generation using random noise. However, we think extending our idea for generation tasks would be an interesting future work to explore.
>
> ---
> **[Q2]  More detailed explanation for the results shown in Figure 2**
>
> Regarding additional explanation for the panorama image experiments, we refer to [GR1] in the common response. Our intention in Figure 2 was to emphasize that our method has more controllability compared to InstructPix2Pix+MultiDiffusion by varying the guidance scale when edit images.
>
> ---
>
> **[Q3]  Reason for the similarly obtained results between SDS and CSD for text-to-3D synthesis**
>
> The reason for the similar results is because of the identical experimental setup (e.g., hyperparameters for NeRF training, see Appendix B.2), especially using the same random seeds. Nonetheless, CSD presents finer details compared to SDS, as shown in Table 3 and Figure 13 in the appendix, illustrating the qualitative benefits of CSD.
>
> ---
>
> **[L1]  Lack of comparison of computation time**
> Following your suggestion, we measure the computation time compared to the baselines in [GR2] in the global response.
>
> ---
>
> **Reference**
>
> [1] Hertz et al., Delta Denoising Score, ICCV 2023
>
> [2] Wang et al., ProlificDreamer: High-Fidelity and Diverse Text-to-3D Generation with Variational Score Distillation, arXiv 2023

---

> > ### Comment · Area_Chair_Utn8 · 2023-08-15
> > **Please check other's reviews and authors' responses**
> >
> > Dear Reviewer yGH2,
> >
> > Could you check the other reviewers' comments and authors' responses?
> >
> > Do you have further questions for the authors?
> >
> > Thanks, Your AC

---

> > > ### Comment · Area_Chair_Utn8 · 2023-08-18
> > > **[AC request] Please check author's responses and make respond ASAP**
> > >
> > > Dear reviewer yGH2,
> > >
> > > As the author-reviewer discussion period is ending soon, could you go over the authors' responses, as well as the questions raised by the other reviewers ASAP?
> > >
> > > Do you have further questions for the authors?
> > >
> > > Thanks, Your AC

---

> > ### Comment · Reviewer_yGH2 · 2023-08-19
> > **Post rebuttal**
> >
> > I want to thank the authors for making an effort in their rebuttal and addressing the reviewers' concerns.
> > The authors responded to most of my concerns. The revised method figure is better and serves the method now.
> >
> > However, I agree that the novelty and the results are not impressive enough, and I suspect that the required revision would be quite significant to address the mentioned clarifications (I still think the N parameter can be confused to be multiple instances of panorama/video/3D scene).
> >
> > Regarding the limited innovation concern raised by other reviewers- if the goal is to address the limitation of SDS, I would ask again why most editing results are presented? It seems that the main contribution of the paper is to claim better editing capabilities over SDS, is that correct?
> >
> > For that reason, I will stay with my current score, and I will make a final decision after a joint discussion with the other reviewers.

---

> > > ### Author Response · Authors · 2023-08-20
> > > **Response to reviewer yGH2**
> > >
> > > Dear reviewer yGH2
> > >
> > > Thank you for your response and we are happy to hear that we have addressed most of your concerns. However, we realized that some of our previous responses (e.g., for your inquiry [Q1]) may cause some confusion, which we would like to clarify in what follows.
> > >
> > > We clarify that our goal is neither `addressing the limitation of SDS’ nor ‘having better editing capabilities over SDS’. Instead, we aim for developing a visual editing method that handles consistency arising in high-dimensional versatile modalities including panorama images, videos, and 3D scenes. To this end, we first formulate the manipulation of high-dimensional versatile modalities as a multi-particle variational inference problem, interpreting the complex visuals as a set of images that satisfy modality-specific consistency. Then, we propose an effective algorithm that adopts SVGD for the diffusion models; importantly, our derivation shows that this adaptation is indeed a generalization of SDS (which is a source for your confusion), but is not just a workaround to overcome the limitation of SDS. Furthermore, we show that providing a better baseline noise, which approximately estimates the score of the source distribution, could improve the editing quality. As recognized by Reviewers v8pz and qUzm, we do believe that our problem formulation and approach are novel, i.e., we address completely a new problem, not for overcoming a limitation of existing approaches. We also remark that although resolving limitations of SDS, e.g., mode collapse, is beyond our scope, generating panoramic images, videos, or 3D scenes using our idea could be an interesting direction to explore in the future.
> > >
> > > Finally, we sincerely appreciate that our manuscript has been improved by incorporating valuable feedback from the reviewers. For instance, following your suggestion, we will include a detailed explanation of the definition of parameter N for different modalities to enhance the presentation. However, we believe that the requested clarifications will not change the essential value of our original paper.
> > >
> > > If you have any further concerns, questions, or suggestions, please do not hesitate to let us know.
> > >
> > > Thank you very much,
> > >
> > > Authors

---

### Official Review · Reviewer_j3eS · 2023-07-05

**Soundness:** 3 good
**Presentation:** 3 good
**Contribution:** 3 good
**Rating:** 5
**Confidence:** 4

**Summary:**

This paper presents a novel method called Collaborative Score Distillation (CSD) for consistent visual synthesis and manipulation. The proposed CSD-Edit utilizes pre-trained text-to-image models and can be competent for panorama image editing, video editing and 3D scene editing tasks, and generate inter-sample consistent results. Sufficient experiments on several tasks have demonstrated that the effectiveness of the proposed method.

**Strengths:**

-	This paper proposes an optimization strategy called Collaborative Score Distillation (CSD) for text-to-image models to perform consistent visual editing, which has demonstrated impressive effectiveness on a variety of tasks including visual editing of panorama images, videos and 3D scenes.
-	The visualization is impressive, qualitatively illustrating the effectiveness of the proposed method and its ability to generate consistent results.
-	The quantitative results also demonstrate the effectiveness of the proposed CSD and its superiority over the existing baseline methods to some extent.


**Weaknesses:**

-	The innovation of the proposed Collaborative Score Distillation is somehow limited, such as combining SDS presented in DreamFusion [26] and SVGD to obtain the general form of SDS.
-	The panorama image editing results tend to produce duplicate content, especially in Figure 2 and Figure 10, and it appears that the diversity of images generated by CD-Edit is limited.
-	The improvement of the numerical results of the quantitative comparison cannot solidly support the superiority of the proposed method, because considering the randomness of the generated/edited results of generative models, it may lead to relatively large fluctuations in the numerical results.
-	Considering that the proposed method is an optimization-based method, the time of optimizing a scene should also be taken into account when compared with other baseline methods. The supplementary materials include the number of iterations of implementation details, but there is still no objective time comparison.
-	Some metrics are expected to assess the intra-image and inter-image consistency, and if objective metrics are lacking, subjective user study may be taken into consideration.


**Questions:**

-	Considering the randomness of the results of the generative models, I’m curious about the criteria used to select the results compared in the paper. This detail should be explained for both the baseline methods and the proposed method to eliminate artificially introduced bias.
-	Repeatability in the generated/edited images may be worth further analysis and ablation study.
-	Other concerns have already been mentioned in Weakness.


**Limitations:**

Yes, the authors have addressed the limitations and potential negative societal impact of their work.

---

> ### Author Rebuttal · Authors · 2023-08-09
>
> Dear reviewer j3eS,
>
> We sincerely appreciate your thoughtful comments, efforts, and time to improve our manuscript. We respond to each of your questions and concerns one-by-one in what follows. Please let us know if you have any comments/concerns that we have not addressed up to your satisfaction.
>
> ---
>
> **[W1]  Limited innovation of proposed method**
>
> As also recognized by Reviewers v8pz and qUzm, we emphasize that our method is not just a naive combination of existing techniques, but rather a novel composition of ideas to address the limitation of SDS in a principled manner. Specifically, we first claim the lack of inter-sample consistency under SDS, which hinders its broader potential applications, e.g., to higher-dimensional visual synthesis. We addressed these challenges by casting them as a multiple-sample variational inference problem and proposed a method that uses SVGD. This reinterpretation serves not only as a simple solution to the problem but also as a practical innovation that is scalable to recent diverse tasks - e.g. adapting text-to-image diffusion models for high-dimensional manipulation. We believe that this could be a useful addition to the field, extending the applicability of SDS for consistent synthesis, which is also highlighted by reviewer v8pz.
>
> ---
>
> **[W2 & Q2]  Lack of diversity and repeatability in the edited results when applying CSD-Edit**
>
> First, we note that our goal is to generate samples consistently in order to demonstrate the power of our method, which allows synchronous editing across a set of images. Nevertheless, it is possible to control the diversity of the generated image with different mini-batch sizes. Note that using a large minibatch size B would encourage more consistency among samples (see our response to reviewer qUzm [Q3] for more explanation). As shown in Figure 3 in the supplementary PDF file, varying the minibatch size B can control the diversity of the generated output: given similar penguins in the source image, the diverse chickens can be generated by using a small batch size. It is important to note that even with this diversity, coherent structures are preserved. This is because randomly selected samples are processed synchronously during each optimization iteration. Lastly, please refer to [Q3] of response to Reviewer qUzm in additional ablation study on the effect of batch size.
>
> ---
>
> **[W3 & Q1]  Consideration of randomness in evaluation**
>
> Since the variance in editing (not generation) tasks from different random seeds is relatively small, we did not highlight the randomness in the evaluation. Instead, we ensured that the hyperparameters were fine-tuned for baseline methods for fair comparison. However, to address your concern, we conducted an additional evaluation with the randomness of generation models into consideration. We repeated under the identical experimental setup using 5 different random seeds and reported average scores along with standard deviations for each method to evaluate desired edits. As demonstrated in Table 1 and Table 2 of the supplementary pdf, CSD-Edit consistently outperforms the baselines, highlighting the robustness of our approach across different runs in achieving desired edits.
>
> ---
>
> **[W4]  Lack of comparison of computation time**
>
> To address your concern, we measure the computation time and compare with the baselines. Please refer to [GR2] in the common response for further details.
>
> ---
>
> **[W5]  Lack of subjective user study**
>
> In addition to our objective evaluation, we conduct additional user studies to compare with video editing and 3D scene editing baselines. Notably, ours outperforms the baselines by a large margin. Please refer to [GR3] in the common response.

---

> > ### Comment · Area_Chair_Utn8 · 2023-08-15
> > **Please check others' review and authors' responses**
> >
> > Dear Reviewer j3eS,
> >
> > Could you check the other reviewers' comments and authors' responses?
> >
> > Do you have further questions for the authors?
> >
> > Thanks, Your AC

---

> > ### Comment · Reviewer_j3eS · 2023-08-16
> >
> > Thanks for the responses to my comments!
> > I have carefully read the comments of other reviewers and the author's responses.
> > This rebuttal has solved most of my puzzles. Although I still think the novelty is not very impressive, the completion and writing of this work are satisfactory. Therefore, I decide to raise my score from 4 to 5.

---

> > > ### Author Response · Authors · 2023-08-16
> > > **Thank you for the response**
> > >
> > > Dear reviewer j3eS
> > >
> > > Thank you for your response! We again sincerely appreciate your efforts and time in reviewing and providing incisive comments on our paper.
> > >
> > > We are pleased to hear that our rebuttal addressed your concerns well. If you have any further concerns, questions, or suggestions, please do not hesitate to let us know.
> > >
> > > Thank you very much!
> > >
> > > Authors

---

### Official Review · Reviewer_qUzm · 2023-07-05

**Soundness:** 3 good
**Presentation:** 3 good
**Contribution:** 3 good
**Rating:** 6
**Confidence:** 4

**Summary:**

The paper presents a novel method, Collaborative Score Distillation (CSD), for diffusion models. The authors propose a new approach to score distillation that leverages the inter-sample relationships to generate more consistent and coherent images. The paper also introduces CSD-Edit, an extension of CSD, which enables the editing of images, videos, and 3D representations. The paper demonstrates the effectiveness of the proposed approach through extensive experiments, showing that the method outperforms existing methods in various tasks, including high-resolution image editing, video editing, and 3D scene editing.

**Strengths:**

(+) The paper presents a novel approach to score distillation that takes into account the inter-sample relationships. This is a departure from existing methods, which typically focus on individual samples. The idea of using Stein Variational Gradient Descent (SVGD) to enforce consistency among samples shows promise in improving the quality of generated images. The ability to edit images, videos, and 3D could have wide-ranging applications in various fields.

(+) The paper is well-written and the proposed methods are clearly explained.

(+) The authors demonstrate the effectiveness of their approach through extensive examples (in the main paper, supplementary document, and webpage).

**Weaknesses:**

(-) The results presented in the paper are not entirely convincing. For instance:
  * In Figure 2, the style editing results between CSD-Edit and Instruct-Pix2Pix + MultiDiffusion are difficult to discern quantitatively. Moreover, noticeable stitching artifacts are visible in the results.
  * The video editing results available on the webpage exhibit a prominent flickering effect, which detracts from the perceived quality of the output.
  * In Table 2, the advantages of CSD-Edit over Instruct-NeRF2NeRF in 3D scene editing are marginal, which raises questions about the practical superiority of the proposed method.

(-) The paper does not provide a robustness analysis of the proposed method. It would be beneficial to understand how the method performs under different conditions or with different types of input.

(-) Please refer to the Questions section for more comments.

**Questions:**

- Could the authors provide more details on the computational efficiency of their method, particularly in comparison with other score distillation methods such as SDS and SJC? It would be beneficial to understand the trade-offs between the quality of the results and the computational resources required.
- How does the method handle errors or biases in the pre-trained Instruct-Pix2Pix model?
- How is the number of samples (N) determined and selected in different experiments and applications (as per Eq. (8))? This appears to be a crucial hyperparameter in the proposed method, and a more detailed explanation of how it is chosen would be helpful.
- In Section 4.4, how are view-dependent prompts and CSD unified and integrated for use in text-to-3D applications?
- The multi-head (or Janus) problem is a well-known issue in text-to-3D applications. Can CSD be used to address this problem? If so, how effective is it, and are there any limitations or challenges in using CSD for this purpose?

**Limitations:**

The authors discuss some limitations of their approach in the supplementary material, but these could be more prominently addressed in the main paper. In particular, the issue of artifacts in high-resolution image editing and the flickering effect in video editing are significant limitations that should be discussed in more detail.

---

> ### Author Rebuttal · Authors · 2023-08-09
>
> Dear reviewer qUzm,
>
> We sincerely appreciate your efforts and comments to improve the manuscript. We respond to your comment in what follows.
>
> ---
>
> **[W1] Some of the results which are not entirely satisfactory**
>
> For your information, we further clarify our experimental results and commit to including these findings in our final manuscript to provide a more comprehensive assessment of our proposed method.
>
> - (Comparison with InstructPix2Pix+MultiDiffusion): The superiority of our method over InstructPix2Pix+MultiDiffusion lies in the controllability over instruction-fidelity across different guidance scales. Please refer to [GR1] in the global response for further details.
>
> - (Regarding the remaining stitching and flickering effects): While our CSD-Edit shows improved consistency compared to prior state-of-the-art baselines across all modalities, this might not be an ultimate approach in ensuring absolute consistency due to the inability of diffusion models and autoencoders. Thus, injecting modality specific data prior could be a promising future work. We will discuss this limitation more prominently in our final manuscript.
>
> - (Quantitative results in 3D scene editing): To address your concerns regarding the quantitative results for 3D scene editing, we conduct additional user studies including other editing tasks. Notably, ours outperforms the baselines by a large margin. Please refer to [GR3] in the global response for further details.
>
> ---
>
>
> **[W2 / Q2] Robust analysis of the proposed method / Errors and bias inherited from pre-trained Instruct-Pix2Pix**
>
> Here, we present a detailed explanation on how our method performs on different conditions or types of input. Since we aim to distill the generative prior of InstructPix2Pix, our method performs well at which InstructPix2Pix excels. For instance, our method shows remarkable performance in stylization or changing multiple objects to others. Also, our method is able to do image editing with multiple prompts, i.e., region-wise edit of a panorama image, in Appendix B.1.
>
> Besides, as we mentioned in the limitation section, the errors and biases of InstructPix2Pix might be transferred to our method. In particular, InstructPix2Pix often edits undesirable objects. Therefore, when editing a panorama image without SVGD, we often observe that unwanted objects are changed in some patches, breaking the consistency of the output image. On the other hand, we empirically observe that our mixing scores in CSD acts as a regularizer that prevents abrupt change in images, ensuring better consistency. For instance, see Figure 4 in the supplementary PDF for visual examples: the tiger is generated in the unwanted region of a source image when SVGD is not applied in update.
>
> ---
>
> **[Q1] Trade-offs between the quality of the results and required computational resources compared with SDS and SJC**
>
> We measure the computation time and compare it with the baselines in [GR2] of the global response.
>
> ---
>
> **[Q3] Detailed explanation about the number of samples N**
>
> For clarification, we denote N as total number of images (e.g., total patches of a panorama image or total frames of a video) and we update B minibatch of samples per iteration. Intuitively, using a large B would encourage more consistency among samples, but it is more computationally expensive. Also, we observed that using large batch sizes dilutes the effect of editing. Thus, the batch size controls the tradeoff between computation time, editing quality, and preserving consistency. To further verify our choice of B, we provide additional ablation study on the panorama image editing experiments. Given the same experimental setup as in Section 4.1 with fixed guidance scale=7.5, we swept over B = 4, 8, 12, and measure the CLIP source-target image similarity, CLIP directional image-text similarity, and computation time (iteration per second for total 200 iterations, measured on single A100 40GB GPU). The following table below demonstrates the effect of batch size.
>
> \begin{array}{lccc}
> \text{Batch size} & \text{CLIP Image Sim.} & \text{CLIP Directional Sim.} & \text{Time (iter / sec)} \newline
> \hline
> \text{B=4}   & 0.6392 & 0.2401 & 2.86  \newline
> \text{B=8}   & 0.6917 & 0.2165 & 1.47  \newline
> \text{B=12} & 0.7394 & 0.1953 &  1.02  \newline
> \end{array}
>
> Also, we provide qualitative examples on the effect of batch size in Figure 3 of supplementary PDF. We show that one can control the diversity of generated output, e.g., same penguins are changed to diverse chickens, by choosing appropriate batch size. We will include this ablation study in our final manuscripts.
>
> ---
>
> **[Q4 & Q5] View-dependent prompting and handling Janus problem in text-to-3D synthesis via CSD**
>
> In Figure 6 of our manuscript, we show that CSD enables consistent visual synthesis of 2D images when view-dependent prompting is applied. While these view-dependent prompts might guide the generation of different shapes or contents for each image, CSD helps to generate coherent objects across different views by updating their scores synchronously. Based on this observation, we apply CSD into text-to-3D synthesis by computing CSD on sampled views at each iteration.
>
> Through experimental results shown in Table 3 and Figure 13 in Appendix, we demonstrate the effectiveness of CSD on text-to-3D generation, especially in potent to mitigate the Janus problem. We believe that the consistent update among views, facilitated by the reduced variance, results in learning better geometry, thus helping better 3D synthesis. Nonetheless, it is important to note that our approach does not entirely remedy the fundamental cause of Janus problem. The fundamental cause of the Janus problem resides in the insufficient understanding of 3D geometry and a lack of diverse views during the training of diffusion models, specifically those views other than front faces.

---

> > ### Comment · Area_Chair_Utn8 · 2023-08-15
> > **Please check others' reviews and authors' responses**
> >
> > Dear Reviewer qUzm,
> >
> > Could you check the other reviewers' comments and authors' responses?
> >
> > Do you have further questions for the authors?
> >
> > Thanks,
> > Your AC

---

> > ### Comment · Reviewer_qUzm · 2023-08-18
> >
> > Dear Authors,
> >
> > Thank you for providing a detailed rebuttal in response to the initial reviews. At this time, I do not have any further questions regarding the paper. I will make my final decision after a collective discussion with the other reviewers.
> >
> > Best,
> >
> > Reviewer qUzm

---

> > > ### Author Response · Authors · 2023-08-19
> > > **Thank you for the response**
> > >
> > > Dear reviewer qUzm
> > >
> > > Thank you for your response! We again sincerely appreciate your efforts and time in reviewing and providing incisive comments on our paper.
> > >
> > > We are pleased to hear that our rebuttal addressed your concerns well. If you have any further concerns, questions, or suggestions, please do not hesitate to let us know.
> > >
> > > Thank you very much!
> > >
> > > Authors

---

### Official Review · Reviewer_v8pz · 2023-07-07

**Soundness:** 4 excellent
**Presentation:** 3 good
**Contribution:** 4 excellent
**Rating:** 8
**Confidence:** 3

**Summary:**

This paper presents a novel method for achieving consistent visual synthesis using a diffusion model. Specifically, the authors extend Score Distillation Sampling to accommodate more complex visual modalities, represented as multiple images. They introduce a principled method to jointly optimize these multiple samples (referred to as particles), ensuring that each sample matches the distribution of an image (evaluated by a pre-trained diffusion model) while maintaining consistency with each other. This methodology is rooted in Stein Variational Gradient Descent. The authors successfully demonstrate a substantial performance improvement in visual quality and consistency across a diverse range of visual editing and synthesis tasks, including panorama editing, video editing, and 3D editing.




**Strengths:**

S1: The proposed method is well-founded and extends the original Score Distillation Sampling (SDS) concept to a novel setting that requires the simultaneous optimization of multiple images. Achieving consistency among different samples is non-trivial and often depends on domain-specific structure (e.g. nerf in DreamFusion, cross attention control in video synthesis). This paper introduces a more principled technique for consistent synthesis, marking a significant contribution to the field.

S2: The overall presentation of the paper is commendable, with a clear structure and explanation that makes it easy to follow.

S3: The method proposed is theoretically sound and delivers impressive results across a wide range of editing tasks. Besides its theoretical contribution, the method demonstrates strong practical applications, incorporating application-specific structure for panorama, 3D, and video. This demonstrates its potential as a solid baseline for future methods in the field.




**Weaknesses:**

While the current approach achieves a degree of consistency by jointly optimizing different images using Stein Variational Gradient Descent, it doesn't necessarily guarantee flawless consistency. This is evidenced by the presence of artifacts at patch boundaries and flickering between video frames. Therefore, consistent synthesis may still necessitate the incorporation of more general structures, such as volumetric rendering, in future visual synthesis pipelines (3D or Video Editing).

**Questions:**

This is not related to my evaluation. But could the author compare this paper with a concurrent paper ProlificDreamer [1]. Specifically, in Section 3.2, the authors mention the need to substitute the random noise with the score predicted by an unconditional model. I am interested in understanding whether this is related to the second noise prediction feature present in ProlificDreamer




[1] Wang, Zhengyi, et al. "ProlificDreamer: High-Fidelity and Diverse Text-to-3D Generation with Variational Score Distillation." arXiv preprint arXiv:2305.16213 (2023).

**Limitations:**

Yes.

---

> ### Author Rebuttal · Authors · 2023-08-09
>
> Dear reviewer v8pz,
>
> We sincerely appreciate your thoughtful comments, efforts, and time to improve our manuscript. We respond to each of your questions and concerns one-by-one in what follows. Please let us know if you have any comments/concerns that we have not addressed up to your satisfaction.
>
> ---
>
> **[W1] How to ensure flawless and complete consistency going further from proper consistency between a set of images?**
>
> At a high level, CSD aims at preserving the consistency between the source images when updating with pre-trained text-to-image diffusion models. In principle, we cast this as a multi-particle variational inference problem and apply Stein
> Variational Gradient Descent algorithm. Through empirical validation, we show that CSD can generate images that follow the text instruction without breaking the consistency. However, we acknowledge that our method might not guarantee the `perfect flawless’ consistency. We also agree with your suggestion in that incorporating modality-specific generators, such as neural fields, or utilizing better priors to ensure consistency, e.g., optical flow for temporal consistency, could be definitely an interesting future work to explore when adapting text-to-image diffusion models to high-dimensional visual synthesis.
>
> ---
>
> **[Q1] Additional comparison to a concurrent work, ProlificDreamer [1] to understand the role of subtracted noise**
>
> Thank you for bringing out the concurrent work, ProlificDreamer, which is very relevant to our paper! Both CSD and ProlificDreamer are based on similar methods, but the tasks of interest are quite different, leading to different choices of second noises for each: we are interested in manipulation of various modalities including images, videos, and 3D scenes, while ProlificDreamer only considers text-to-3D generation. In its basis, CSD and ProlificDreamer have identical objectives in distilling the generative prior of pre-trained text-to-image diffusion models using weighted KL divergence (i.e., Eq.(4) in our manuscript). Then the particle-based optimization using Wasserstein gradient flow [2] is given by the different between the score function of target distribution $p$ and source distribution $q$:
> $$d\mathbf{x} = (\nabla_{\mathbf{x}} \log p(\mathbf{x}) - \nabla_{\mathbf{x}} \log q_t(\mathbf{x}))dt $$
> Here, the score of target distribution is given by a pre-trained diffusion model, while the score function of source distribution is not present as default. In our paper, we use the image-conditional noise estimate of InstructPix2Pix, which approximately estimates the score of source distribution. In ProlificDreamer, they resort to online fine-tuning of a diffusion model with respect to the current NeRF scenes. Notably, we additionally compute kernels and mix the scores of multiple samples, which enhances consistency. We believe that ProlificDreamer and CSD are complementary works that the idea of CSD can be used in ProlificDreamer to enhance 3D consistency in text-to-3D generation, which we leave as future work.
>
> ---
> **Reference**
>
>
> [1] Wang et al., ProlificDreamer: High-Fidelity and Diverse Text-to-3D Generation with Variational Score Distillation, arXiv 2023
>
> [2] Richard Jordan, David Kinderlehrer, and Felix Otto. “The variational formulation of the Fokker–Planck equation”. In: SIAM journal on mathematical analysis 29.1 (1998). Publisher: SIAM, pp. 1–17.

---

> > ### Comment · Area_Chair_Utn8 · 2023-08-15
> > **Please check others' review and authors' responses**
> >
> > Dear Reviewer v8pz,
> >
> > Could you check the other reviewers' comments and authors' responses?
> >
> > Do you have further questions for the authors?
> >
> > Thanks,
> > Your AC

---

> > ### Comment · Reviewer_v8pz · 2023-08-17
> > **Reply to Response**
> >
> > Thank you for the response! It addressed my original questions. Other reviewer's concerns about the inference efficiency and qualitative results are still valid. I will finalized the rating based on discussions with other reviewers.

---

> > > ### Author Response · Authors · 2023-08-19
> > > **Thank you for the response**
> > >
> > > Dear reviewer v8pz
> > >
> > > Thank you for your response and we are pleased that we have addressed your concerns.
> > >
> > > Furthermore, we would like to clarify that our method can achieve better computational efficiency compared to baselines while maintaining the superior editing quality simultaneously. We may have inadvertently given the impression that our method is less computationally efficient in our original submission, as computation time was not our primary focus.
> > >
> > > However, this is not true: one can easily enhance its computational efficiency by reducing the total optimization iterations and utilizing a higher learning rate, allowing for more edits in each iteration. To further clarify this, we have conducted additional experiments focusing on both computational efficiency and editing quality. Please refer to [GR2-1] in the global response for more details.
> > >
> > > In addition, regarding the qualitative results, could you provide more detailed information if possible? Your insights are invaluable, and giving further feedback on them will greatly help us to strengthen our manuscript. We believe that we thoroughly addressed the concerns about the qualitative results raised by other reviewers (qUzm, j3eS). Nonetheless, we are willing to address any remaining concerns you might have.
> > >
> > > Once again, we deeply appreciate your time and efforts on our paper.
> > >
> > > Sincerely,
> > >
> > > Authors

---

> > > > ### Comment · Reviewer_v8pz · 2023-08-19
> > > > **Reply to Author**
> > > >
> > > > Dear Authors,
> > > >
> > > > Thank you for the further clarifications. I have read through other reviewers comments and the new responses. My concern regarding the computational efficiency and Figure 2 are addressed. Multiple presentation clarifications raised by other reviewers are good points to revise in the final version.
> > > >
> > > > Additionally, I agree with reviewer yGH2 that the current method doesn't support generation well as it still suffers from the mode collapse issue of SDS (and the repulsive kernel on image space is not sufficient for preventing mode collapse as we can get samples that are semantically similar despite large pixel-wise distance). I would suggest the authors to discuss this limitation further.
> > > >
> > > > Overall, I am still positive about the submission and remain my accept rating.

---

> > > > > ### Author Response · Authors · 2023-08-20
> > > > > **Response to reviewer v8pz**
> > > > >
> > > > > Thank you for your valuable feedback! We are glad to see that your concerns have been addressed.
> > > > >
> > > > > Although our main interest lies in the editing (not generation) of panoramic images, videos, or 3D scenes, we believe that CSD has the potential to be used in their generation. As presented in Section 4.4 of our manuscript, we have done some experiments on text-to-3D generation to show how CSD can improve generation performance over SDS. In particular, we showed the effect of collaborative score distillation in improving the geometry and quality of text-to-3D generation, while resolving other limitations of SDS, e.g., mode collapse, was beyond our scope.
> > > > >
> > > > > Nevertheless, as we mentioned in our initial response, subtracting different baseline noises may be a key idea in solving the mode collapse problem of SDS (which was also investigated in the ProlificDreamer paper). It is an interesting research topic and we leave it for future work.
> > > > >
> > > > > We will also include this discussion in the Limitations and Future Work sections.
> > > > >
> > > > > Thank you very much,
> > > > >
> > > > > Authors

---

### Author Rebuttal · Authors · 2023-08-09

Dear reviewers and AC,

We sincerely appreciate your valuable time and effort spent reviewing our manuscript.

As reviewers highlighted, we believe our paper presents a principled and novel method (v8pz, qUzm) that performs effective visual editing of versatile modalities (v8pz, qUzm, j3es, yGH2), validated by extensive experiments in both quantitative and qualitatively (v8pz, qUzm, j3es), followed by comprehensive presentation (v8pz, qUzm, yGH2).

Here, we collected the common questions that multiple reviewers have asked and responded to each question one-by-one in what follows. We also kindly ask you to check out the attached supplementary PDF file together. Please let us know if you have any comments/concerns that we have not addressed up to your satisfaction.

---

**[GR1] More detailed explanation for the results shown in Figure 2 (Reviewer qUzm, yGH2)**

Figure 2 illustrates two benefits of our method: spatial consistency and instruction-fidelity. Patch-wise editing of a panorama image results in spatial inconsistencies due to visible patch boundaries (Figure 2, right top). This is mitigated in InstructPix2Pix+MultiDiffusion (Figure 2, middle row) by using overlapping patches, but the edited image loses its fidelity to the instruction as the scores are diluted by other scores, e.g., one patch may respond to the instruction much more or much less compared to others, thus the effect is diluted in such cases. On the other hand, CSD is able to mitigate such a diluting effect by optimizing with a subset of images. Thus, in Figure 2, given the same guidance scale, our method shows better fidelity to the instruction.

The effect of guidance scales on the instruction-fidelity of image editing is demonstrated in Figure 5 of our manuscript. Each dot on the graph represents different guidance scales, and a direct comparison of each guidance scale shows a noticeable gap between InstructPix2Pix+MultiDiffusion and CSD-Edit in terms of CLIP directional scores. This underscores the superiority of our method in balancing source-target image consistency and instruction-fidelity across scales, thereby highlighting its controllability, particularly given the subjective nature of achieving a desired edit. In the enhancement of representation, we revise Figure 5 (now Figure 2 in the supplementary PDF) following the reviewers’ comments and will include the revised figure along with a detailed explanation in our final manuscript.

***
**[GR2] Measurement and comparison of computation time (Reviewer qUzm, j3eS, yGH2)**

Regarding the computational efficiency of our method, we measure the computation times and compare them with the baselines of each task. All these evaluations were conducted on a single NVIDIA A100 80GB and AMD EPYC 7V13 64-Core Processor.

- For panorama image editing experiments, we compare with the baseline InstructPix2Pix+MultiDiffusion. Note that the computation time of both methods depends on the input image resolution, whereas we show that our method becomes more efficient as the resolution goes higher. The following table below shows how the computation time (total time in seconds) differs by the size of the input image.
 \begin{array}{lccc}
\text{Method} & \text{Resolution} & \text{Total time (sec.)} \newline
\hline
 \text{InstructPix2Pix+MultiDiffusion} & 1920\times640 & 62  \newline
\text{CSD-Edit (Ours)} & 1920\times640 & 68  \newline
\hline
 \text{InstructPix2Pix+MultiDiffusion} & 3968\times 4352  & 487  \newline
\text{CSD-Edit (Ours)} & 3968\times 4352  & 275   \newline
\end{array}

  Note that the baseline method requires computing noise estimates of every patch at each diffusion step, while our method only requires computing a minibatch of patches per iteration.

- In video editing experiments, we measure the total computation time of our method and baseline methods to obtain the results shown in Figure 11. The results are shown in the following table below:
\begin{array}{lc}
\text{Method} & \text{Total time (sec.)} \newline
\hline
\text{FateZero}   & 192 \newline
\text{Pix2Video}   & 77  \newline
\text{CSD-Edit (Ours)}   & 423  \newline
\end{array}

  Although our method requires more computation time compared to the baselines, users can choose to early stop the optimization when they achieve a desired edit, which is not an attribute of the baseline methods as they resort to diffusion samplers.

- In text-to-3D generation experiments, when directly comparing with DreamFusion (or SJC), there is a slight increase in the computational cost due to the usage of LPIPS metric as distance for RBF kernel. For instance, it takes 1 hour to generate a 3D model with DreamFusion, while using CSD takes 84 minutes. We note that the increment is not substantial, however, ours yields better qualities with finer details, as shown in Table 3 and Figure 13 in the Appendix of our manuscript.

We will add this information to our final manuscript.

---

**[GR3] User study (Reviwer qUzm, j3eS)**

While we have primarily relied on objective metrics for assessing consistency and instruction-fidelity, we agree that a subjective user study could provide valuable insights, especially given the subjective nature of editing tasks. Thus, we conducted additional user studies, where we asked three questions in evaluating the editing methods: the consistency of the edited results, frame-wise image instruction-fidelity, and the editing quality.  For each of the three studies, we asked 20 subjects to rank different methods. As shown in Table 3 and Table 4 in the supplementary PDF file, our method consistently outperforms others achieving the best user preferences across all three aspects. We commit to including these findings in the final version of our manuscript to provide a more comprehensive assessment of our proposed method.

---

> ### Author Response · Authors · 2023-08-19
>
> **[GR2-1] Computational efficiency compared to baselines**
>
> We would like to clarify that our method can obtain better computational efficiency in comparison with the baselines while still outperforming them in editing quality. Indeed, the computational efficiency of our method, CSD-Edit, on video editing can be easily improved by reducing the number of iterations for optimization while using a larger learning rate. The results are shown in the table below:
>
> \begin{array}{l | c | c | c | c}
> \text{Method} & \text{Total time (sec.)}  \downarrow & \text{CLIP Directional Similarity} \uparrow & \text{CLIP Image Consistency} \uparrow & \text{LPIPS } \downarrow \newline
> \hline
> \text{FateZero}   & 192  & 0.312 \small{\pm0.003}& 0.948 \small{\pm0.001} & 0.264 \small{\pm0.002}\newline
> \text{Pix2Video}   & 77 & 0.229 \small{\pm0.001} & 0.948 \small{\pm0.001} & 0.282 \small{\pm0.001}\newline
> \hline
> \text{CSD-Edit (Ours)}   & \mathbf{59}  & \mathbf{0.320 \small{\pm0.001}} & 0.955 \small{\pm0.001} &  0.243 \small{\pm0.001} \newline
> \text{CSD-Edit (Ours)}   & 423  & \mathbf{0.319 \small{\pm0.002}} & \mathbf{0.957 \small{\pm0.001}} & \mathbf{0.235 \small{\pm0.001}} \newline
> \end{array}
>
> As shown in the table, our method, CSD-Edit, consistently outperforms the baselines, even taking the shortest time in editing ($\times$3.3 times faster than FateZero, $\times$1.3 times faster than Pix2Video). Here, using a higher learning rate changes frames more radically, thus slightly degrading the frame consistency, however, the effect is meager as shown in CLIP Image consistency and LPIPS metrics.
>
> Note that in our original manuscript, we put no effort to reduce the computational time since it is a design choice in our framework; the users can choose to early stop the optimization when they achieve a desired edit. Due to this, we rather focused on the superior controllability and quality of our method. Nonetheless, as we demonstrated in [Global Response 2], in panorama image editing experiments, our method already shows similar computation time compared to the baseline, even achieving better compute-efficiency when the image resolution becomes higher. In addition, in video editing experiments, our method not only shows much better compute-efficiency as mentioned above but also maintains superior editing quality compared to baselines.
>
> We will update the final manuscript with these additional experimental results for further clarification.

---

### Decision · Program_Chairs · 2023-09-21

**Decision:**

Accept (poster)

**Comment:**

This paper proposes a collaborative score distillation technique for visual editing, including panorama image, video, and 3D. In addition to editing, the authors also show the results of text-to-3D generation. After the rebuttal, most reviewer concerns are addressed by the authors. However, the reviewers keep the concern about the novelty and the scope of this paper:
1. Leveraging SVGD for the task seems somewhat incremental;
2. It is unclear if the goal is to improve general SDS method, if so, there is no major comparisons on the generation side;
3. All the experiments should compare the proposed approach with the original SDS approach.

After going over the literature and the comments made by the reviewers and the authors, the AC decide to accept the paper as:
1. Many reviewers still hold positive opinions although the novelty may be somewhat incremental.
2. In the rebuttal, the authors specify that the scope of this paper is to propose an effective approach for visual editing, not improving general SDS.
3. The authors do show the comparison to the original SDS approach in some experiments, and in the other experiments, the authors present the results of stronger baselines.

Despite acceptance, the AC sincerely **ask** the authors to
1. Include all the discussions/experimental results in the rebuttal to the paper or supplementary materials.
2. Revise the paper (please consider revising the title as well) to ensure that **the manuscript describes the proper scope**. Specifically, as discussed during the rebuttal, the authors should be responsible to make sure the reader understand that the goal is for "editing" not general "synthesis", e.g., the title is not accurate from this perspective.
3. The authors should include the results of vanilla SDS in all the other experiments presented in the paper.